# ONE CANNOT STAND FOR EVERYONE! LEVERAGING MULTIPLE USER SIMULATORS TO TRAIN TASK-ORIENTED DIALOGUE SYSTEMS

## ABSTRACT

User simulators are agents designed to imitate human users; recent advances have found that Task-oriented Dialogue (ToD) systems optimized toward a user simulator could better satisfy the need of human users. However, this might result in a sub-optimal ToD system if it is tailored to only one *ad hoc* user simulator, since human users can behave differently. In this paper, we propose a framework called MUST to optimize ToD systems via leveraging **M**ultiple **U**ser **S**imula**T**ors.

The main challenges of implementing the MUST fall in 1) how to adaptively specify which user simulator to interact with the ToD system at each optimization step, since the ToD system might be over-fitted to some specific user simulators, and simultaneously under-fitted to some others; 2) how to avoid catastrophic forgetting of the adaption for a simulator that is not selected for several consecutive optimization steps. To tackle these challenges, we formulate MUST as a Multi-armed bandits (MAB) problem and provide a method called MUST$_{\text{adaptive}}$ that balances *i*) the *boosting adaption* for adaptive interactions between different user simulators and the ToD system and *ii*) the *uniform adaption* to avoid the catastrophic forgetting issue. With both automatic evaluations and human evaluations, our experimental results on the restaurant search task from MultiWOZ show that the dialogue system trained by our proposed MUST achieves a better performance than those trained by any single user simulator. It also has a better generalization ability when testing with unseen user simulators. **Moreover, our method MUST$_{\text{adaptive}}$ can efficiently leverage multiple user simulators to train the ToD system by our visualization analysis on convergence speeds.**

## 1 INTRODUCTION

Task-oriented dialogue systems aim to help users accomplish their various tasks (e.g., restaurant reservations) through natural language conversations. Training task-oriented dialogue systems in supervised learning (SL) approaches often requires a large amount of expert-labeled dialogues, however collecting these dialogues is usually expensive and time-consuming. Moreover, even with a large amount of dialogue data, some dialogue states may not be explored sufficiently for dialogue systems [1] (Li et al., 2016b). To this end, many researchers try to build user simulators to mimic human users for generating reasonable and natural conversations. By using a user simulator and sampled user goals, we can train the dialogue system from scratch with reinforcement learning (RL) algorithms. Previous works tend to design better user simulator models (Schatzmann et al., 2007; Asri et al., 2016; Gur et al., 2018; Kreyssig et al., 2018; Lin et al., 2021). Especially, Shi et al. (2019) builds various user simulators and analyzes the behavior of each user simulator in the popular restaurant search task from MultiWOZ (Budzianowski et al., 2018).

In real application scenarios, the deployed dialogue system needs to face various types of human users. A single *ad hoc* user simulator can only represent one or a group of users, while other users might be under-represented. Instead of choosing the best-performing one from many dialogue systems trained by different single user simulators, we believe that it is worth trying to train a dialogue system by leveraging all user simulators simultaneously.

---

[1] We use the dialogue systems to refer to the task-oriented dialogue systems for simplicity in this paper.

In this paper, we propose a framework called MUST to utilize **M**ultiple **U**ser **S**imula**T**ors simultaneously to obtain a better system agent. There exist several simple ways to implement the MUST framework, including a merging strategy, a continual reinforcement learning (CRL) strategy, and a uniform adaption strategy, denoted as $\text{MUST}_{\text{merging}}$, $\text{MUST}_{\text{CRL}}$, and $\text{MUST}_{\text{uniform}}$ respectively (See Sec. 3.2). However, none of them could effectively tackle the challenges: 1) how to efficiently leverage multiple user simulators when training the dialogue system since the system might be easily over-fitted to some specific user simulators and simultaneously under-fitted to some others, and 2) it should avoid a catastrophic forgetting issue. To tackle them effectively, we first formulate the problem as a Multi-armed bandits (MAB) problem (Auer et al., 2002); similar to the exploitation vs exploration trade-off, specifying multiple user simulators should trade off a boosting adaption (tackling the challenge 1) and a uniform adaption (tackling the challenge 2), see Sec. 4.1 for more details. Then we implement a new method called $\text{MUST}_{\text{adaptive}}$ which utilizes an adaptively-updated distribution among all user simulators to sample them to train the dialogue system in the RL training.

Our experimental results on the restaurant search task from MultiWOZ with both automatic evaluations and human evaluations show that the dialogue system trained by our proposed MUST achieves a better performance than those trained by any single user simulator. It also has a better generalization ability when testing with unseen user simulators and is more robust to the diversity of user simulators. **Moreover, the visualization analysis on convergence speeds demonstrates that our $\text{MUST}_{\text{adaptive}}$ is more efficient than $\text{MUST}_{\text{uniform}}$ to leverage multiple user simulators to train dialogue systems**.

Our contributions are three-fold: (1) To the best of our knowledge, our proposed MUST is the first developed work to improve the dialogue system by using multiple user simulators simultaneously; (2) We design several ways to implement the MUST. Especially, we formulate MUST as a Multi-armed bandits (MAB) problem, based on which we provide a novel method $\text{MUST}_{\text{adaptive}}$; and (3) The results show that dialogue systems trained with MUST consistently outperform those trained with a single user simulator through automatic and human evaluations. Especially, it largely improves the performance of the dialogue system tested on out-of-domain evaluation. Furthermore, training the system with the proposed $\text{MUST}_{\text{adaptive}}$ can converge faster than with $\text{MUST}_{\text{uniform}}$.

## 2 BACKGROUND

**Dialogue System.** Task-oriented dialogue systems aim to help users accomplish various tasks such as restaurant reservations through natural language conversations. Researchers usually divide the task-oriented dialogue systems into four modules (Wen et al., 2017; Ham et al., 2020; Peng et al., 2021): Natural Language Understanding (NLU) (Liu & Lane, 2016) that first comprehends user's intents and extracts the slots-values pairs, Dialog State Tracker (DST) (Williams et al., 2013) that tracks the values of slots, Dialog Policy Learning (POL) (Peng et al., 2017; 2018) that decides the dialog actions, and Natural Language Generation (NLG) (Wen et al., 2015; Peng et al., 2020) that translates the dialog actions into a natural-language form. The DST module and the POL module usually are collectively referred to as the dialogue manager (DM) (Chen et al., 2017). These different modules can be trained independently or jointly in an end-to-end manner (Wen et al., 2017; Liu & Lane, 2018; Ham et al., 2020; Peng et al., 2021; Hosseini-Asl et al., 2020).

**User Simulator.** The user simulator is also an agent but plays a user role. Different from dialogue systems, the user agent has a goal describing a target entity (e.g., a restaurant at a specific location) and should express its goal completely in an organized way by interacting with the system agent (Takanobu et al., 2020). Therefore, besides the modules of NLU, DM, and NLG like dialogue systems, the user agent should have another module called Goal Generator (Kreyssig et al., 2018), which is responsible for generating the user's goal. Building a user simulator usually could use an agenda-based approach (Schatzmann et al., 2007; Schatzmann & Young, 2009) designing handcrafted rules to mimic user behaviors or a model-based approach such as neural networks (Asri et al., 2016; Kreyssig et al., 2018; Gur et al., 2018) learned on a corpus of dialogues.

**Training Dialogue Systems with a User Simulator.** At the beginning of a dialogue, the user agent obtains its initial goal from the Goal Generator and then expresses its goal in natural languages. For the system agent, it does not know the user's goal and it should gradually understand the user's utterances, query the database to find entities, and provide useful information to see if it is accomplishing

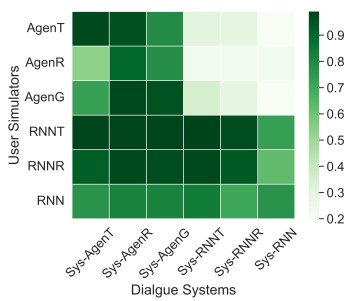

(a) Success rates of different systems.

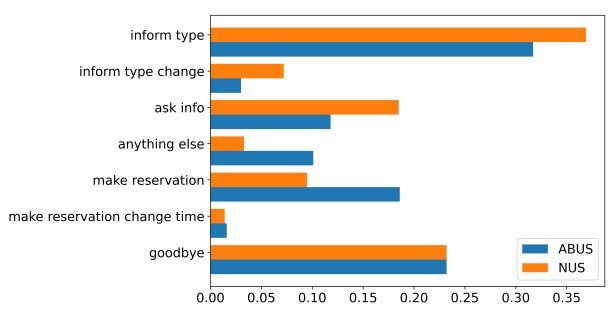

(b) Dialog act distributions of different user simulators.

Figure 1: (a) is the heat map on the success rates of system agents tested by different user simulators on 200 dialogues. (b) shows the dialog act distributions of Agenda-based User Simulators (ABUS) and Neural networks-based User Simulators (NUS) provided by Shi et al. (2019). There exist seven user dialog acts annotated in the restaurant search task from MultiWOZ, as shown on the Y-axis.

the user's task. Since only the system can access the database, the user does not know if its goal can be satisfied. When the database result returned by the system agent is empty, the user agent should learn to compromise and change its goal with the help of Goal Generator (Kreyssig et al., 2018). When the dialogue ends, the user simulator will reward the system agent according to if the system agent accomplishes the task. Then we could use the reward to update the system agent with RL algorithms (Tseng et al., 2021).

## 3 MUST: LEVERAGE MULTIPLE USER SIMULATORS

### 3.1 MOTIVATIONS TO LEVERAGE MULTIPLE USER SIMULATORS

**User simulators behave differently.** Shi et al. (2019) implement six user simulators (AgenT, AgenR, AgenG, RNNT, RNNR, RNN [2]) with both agenda-based methods and neural networks-based methods on the popular restaurant search task from MultiWOZ (Budzianowski et al., 2018). From their experiments, we observed that the dialogue systems trained by different user simulators vary in their performances (i.e., the success rates tested by the same user simulators). For example, when interacting with the user simulator of AgenT, the success rates of the system agents trained by Agenda-based user simulators (i.e., AgenT, AgenR, AgenG) are much higher than the system agents trained by RNN-based user simulators (i.e., RNNT, RNNR, RNN), see Fig. 1(a). The reason might be that these user simulators (i.e., with either handcrafted rules or data-driven learning in their DM modules) have different user dialog act distributions [3] (see Fig. 1(b)) which determines the dialogue state space explored by the dialogue system.

**One cannot stand for everyone.** Users might behave differently, one could design different user simulators with specific user dialog act distributions, see Shi et al. (2019). A single user simulator learned on a task-oriented dialogue corpus can just represent one or a group of users, while the dialogue system needs to accomplish tasks from various human users in real scenarios. We argue that it is beneficial to utilize all different user simulators to train the dialogue system. By leveraging multiple user simulators that have different user dialog act distributions, the dialogue systems can explore a larger dialogue state space, which might improve the ability of the learned dialogue system.

### 3.2 SOME PRELIMINARY PROPOSALS FOR MUST

We propose a framework called MUST, the core idea of which is to train a better dialogue system by leveraging **M**ultiple **U**ser **S**imula**T**ors simultaneously. There are several simple ways to imple-

---

[2]Here we rename the user simulators of SLT, SLR, and SLE in Shi et al. (2019) as RNNT, RNNR, RNN for emphasizing the model structure of their DM modules.

[3]The dialogue policy learning module is essential in both dialogue systems and user simulators. A policy module corresponds to a dialog act distribution since it decides to take which dialog act to respond to the current dialogue state. The user dialog act distribution behind a user simulator determines the diversity of the dialogue state space explored by dialogue systems; therefore it might affect the performance of system agents.

Table 1: The comparison of different strategies for leveraging multiple user simulators.

| | dynamic adaption | avoid catastrophic forgetting | efficiency |
|---|:---:|:---:|:---:|
| $\text{MUST}_{\text{merging}}$ | ✗ | ✗ | ✗ |
| $\text{MUST}_{\text{CRL}}$ | ✗ | ✗ | ✗ |
| $\text{MUST}_{\text{uniform}}$ | ✗ | ✓ | ✗ |
| $\text{MUST}_{\text{adaptive}}$ | ✓ | ✓ | ✓ |

ment our MUST, including a *merging* strategy denoted as $\text{MUST}_{\text{merging}}$, a *Continual Reinforcement Learning* strategy denoted as $\text{MUST}_{\text{CRL}}$, and a *uniform adaption* strategy denoted as $\text{MUST}_{\text{uniform}}$.

**(I) $\text{MUST}_{\text{merging}}$** first samples some dialogues from each user simulator and the corresponding dialogue system trained by this simulator. Then it combines the collected dialogues to train a new user simulator for ensembling different user dialog act distributions. Finally, it uses this new user simulator to train the dialogue system with RL.

**(II) $\text{MUST}_{\text{CRL}}$** [4] treats each user simulator as an independent RL environment. It moves the trained system agent to another environment (i.e., let the system agent interact with another user simulator) if the system converges in the current RL environment.

**(III) $\text{MUST}_{\text{uniform}}$** allows the system agent have chances to interact with all user simulators simultaneously. Different from $\text{MUST}_{\text{CRL}}$, $\text{MUST}_{\text{uniform}}$ puts all user simulators in a single RL environment and adopts the simplest way to specify different user simulators to train the dialogue system, which is to pick a user simulator among all user simulators with a uniform distribution for each iteration in the RL training.

**Challenges to leverage multiple user simulators.** The problem with $\text{MUST}_{\text{merging}}$ is that it becomes difficult to adjust the weights of each user simulator adaptively in the training process. Since the proportions of dialogues from each user simulator are fixed in $\text{MUST}_{\text{merging}}$, some user simulators might be well-adapted and some might not. The $\text{MUST}_{\text{CRL}}$ strategy has a problem of catastrophic forgetting (Khetarpal et al., 2020) and would be sensitive to the order of different user agents interacting with the dialogue system, which might result in obtaining a sub-optimal dialogue system. As Shi et al. (2019) shows, the system agents trained by different user simulators have different convergence speeds and converged performances. Namely, the system agent might be easily fitted to some user simulators but might be hardly fitted to others. **The uniform distribution for user simulator selection under $\text{MUST}_{\text{uniform}}$ will result in inefficient training since it would be redundant to assign the same training costs to let the dialogue system interact with easily-adapted user simulators.** Overall, the challenging problems under the MUST framework are 1) how to efficiently leverage multiple user simulators to train the system agent, and 2) avoiding the catastrophic forgetting issue.

## 4 MUST AS A MULTI-ARMED BANDIT PROBLEM

To tackle the challenges under the MUST, we first formulate MUST as a Multi-armed bandit (MAB) problem, see Sec. 4.1. In Sec. 4.2, we propose a method called $\text{MUST}_{\text{adaptive}}$ to use an adaptively-updated distribution to replace the uniform distribution under the $\text{MUST}_{\text{uniform}}$ for accelerating the MUST training. We briefly compare these different implementations of MUST in Tab. 1.

### 4.1 FORMULATE MUST AS A MULTI-ARMED BANDIT PROBLEM

Adaptively specifying different user simulators to train the dialogue system reminds us of a similar concept in machine learning, which is the *boosting* strategy (Zhou, 2012). From a *boosting* point of view, one should increase the weights of weakly-performing data examples and decrease the weights for well-performing ones. In MUST, we accordingly assume that **it should reduce the interactions between the dialogue system and those user simulators that the system has performed well and increase the interactions between the system and other user simulators that the system performs poorly**. We refer to this strategy as *boosting adaption*.

---

[4]Continual Reinforcement Learning (CRL) Khetarpal et al. (2020) is a sequential learning paradigm for training an agent with RL algorithms.

Meanwhile, we should also give some chances to all user simulators to relieve the catastrophic forgetting issue. We refer to this as *uniform adaption*. Such a trade-off between *boosting adaption* and *uniform adaption* is similar to the *the exploitation vs exploration trade-off* existing in the Multi-armed bandit (MAB) problem (Auer et al., 2002).

Here, we interpret MUST as a MAB problem. We treat each user simulator as an arm. Suppose there are $K$ arms (simulators), and each arm $i$ has a fixed but unknown reward distribution $R_i$ with an expectation $\mu_i$. At each time step $t = 1, 2, ..., T$, one must choose one of these $K$ arms. We denote the arm pulled at time step $t$ as $i_t \in \{1, ..., K\}$. After pulling an arm, it will receive a reward $x_{i_t}$ drawn from the arm's underlying reward distribution. The decision maker's objective is to maximize the cumulative expected reward over the time horizon

$$\sum_{t=1}^{T} \mathbb{E}[x_{i_t}] = \sum_{t=1}^{T} \mu_{i_t}. \tag{1}$$

In MUST, the reward received in each arm-pulling step refers to the possible performance gain of the dialogue system after it interacts with a selected user simulator. A significant *difference* between the standard MAB problem and MUST is that the reward expectation of a user simulator (arm) in MUST is not static; it changes over time. For example, by consecutively interacting with the same user simulator, the performance gain (reward) of the system will decay since the system might be in saturation or overfitting to this simulator. Moreover, the performance gain of the system after interacting with a simulator might increases if the simulator has not been selected for a period. To deal with this *difference*, we should tailor the solution of MAB to the MUST framework.

## 4.2 TRAINING THE DIALOGUE SYSTEM WITH MUST$_{\text{adaptive}}$

To solve this MAB problem in MUST, we implement a method called MUST$_{\text{adaptive}}$ with a two-phase procedure, as presented in Algorithm 1. Similar to the UCB1 [5] algorithm, MUST$_{\text{adaptive}}$ specifies user simulators in a uniform distribution to train the dialogue system $S$ in the first $T_{\text{warmup}}$ steps (i.e., in the *warm-up phase*). After that, the *adaptive phase* will balance the boosting adaption and the uniform adaption by introducing an adaptively-updated distribution $\boldsymbol{p}$, which is used to specify different user simulators to train the system $S$ in later RL training. To accelerate the RL training, intuitively, $\boldsymbol{p}$ is expected to *assign lower weights to user simulators with which $S$ already performs well* and *higher weights to those user simulators with which $S$ performs poorly*.

(1) **Warm-up phase**: in the first $T_{\text{warmup}}$ dialogues, we use a uniform distribution to sample all user simulators to train the system agent $S$ (lines 4-7). This phase is mainly used to warm up the dialogue system $S$ and make it have little ability to converse with all user simulators.

(2) **Adaptive phase**: in this phase, the distribution $\boldsymbol{p}$ used to sample all user simulators will be adaptively updated, which is why we call this phase **adaptive phase**. When this phase begins (i.e., $t = 0$), we will first evaluate the performance (i.e., the success rate $\bar{x}_j, j \in \{1, \cdots, K\}$) of the dialogue system $S$ trained after the **warm-up phase**. The success rate $\bar{x}_j$ is obtained by letting $S$ interact $d$ times with the simulator $U_j$ (e.g., $j \in \{1, ..., K\}$) and calculating the success rates.

Inspired by UCB1 Auer et al. (2002), we design a calibrated **performance expectation** $\hat{x}_j$ of the system agent $S$ interacting with each user simulator $U_j$ **taking exploration into consideration beyond pure exploitation:**

$$\hat{x}_j = \underbrace{\bar{x}_j}_{\text{exploitation}} + \underbrace{\sqrt{\frac{2 \ln t}{T_{j,t}}}}_{\text{exploration}}, j \in \{1, ..., K\}; \tag{2}$$

where $\bar{x}_j$ is the success rate of the system agent $S$ tested with user simulator $U_j$, $T_{j,t}$ is the number of times user simulator $U_j$ has been selected with so far. Then we normalize $\hat{x}_j$ into

$$z_j = 1/(\hat{x}_j - \tau \min(\{\bar{x}_1, \cdots, \bar{x}_K\})), \tag{3}$$

---

[5]**There exists an algorithm called the Upper Confidence Bound 1 (UCB1) (Auer et al., 2002) that could solve the MAB problem. This policy first pulls each arm once in the first $K$ steps, then will play the arm that could maximize the sum of two terms: $i_t = \arg\max_i \left( \bar{x}_i + \sqrt{\frac{2 \ln t}{T_{i,t}}} \right)$ from $t = K + 1$ to $T$.**

---

**Algorithm 1:** Implement MUST$_\text{adaptive}$ with the *modified* UCB1 algorithm

---

**Input:** K fixed **U**ser simulators $\mathbf{U} = \{U_1, U_2, \cdots U_K\}$ and the values of hyperparameters
      $T_\text{warmup}, T, e, d, \tau$;

1   **Initialization**: randomly initialize **S**ystem agent $S$;
2   **Initialization**: initialize the simulator sampling distribution $\boldsymbol{p}$ as a uniform distribution.
3   (1) **Warm-up phase:**
4   **for** $t = 0, ..., T_{warmup} - 1$ **do**
5      sample a simulator $U_j$ in **U** w.r.t. the distribution $\boldsymbol{p}$;
6      synthesize a new dialogue using the system agent $S$ and the sampled $U_j$ ;
7      use the reward obtained for the dialogue to update $S$ with a RL algorithm;

8   (2) **Adaptive phase:**
9   **for** $t = 0, ..., T - 1$ **do**
10      **if** $t\%e == 0$ **then**
11         **for** $j = 1, ..., K$ **do**
12            evaluate the performance i.e. the success rate $\bar{x}_j$ of the agent $S$ by letting it interact
               $d$ times with the simulator $U_j$;
13         update $\boldsymbol{p}$ based on these success rates $\{\bar{x}_1, ..., \bar{x}_K\}$ (see Eq. 2, Eq. 3, and Eq. 4);
14      **else**
15         sample a simulator $U_j$ in **U** w.r.t. the distribution $\boldsymbol{p}$;
16         synthesizing a new dialogue using the system agent $S$ and the sampled $U_j$ ;
17         use the reward obtained for the dialogue to update $S$ with a RL algorithm;

**Output:** The learned dialogue system $S$.

---

Eq. 3 penalizes the user simulators with which the dialogue system already performs well in the expectation term. Where the hyperparameter $\tau$ is the smooth factor for distribution $\boldsymbol{p} = \{\boldsymbol{p}_1, \cdots, \boldsymbol{p}_K\}$ – the larger $\tau$ is, the sharper $\boldsymbol{p}$ is. Each probability $\boldsymbol{p}_j$ in $\boldsymbol{p}$ is calculated as

$$\boldsymbol{p}_j = \frac{z_j}{\sum_{i=1}^{K} z_i}. \tag{4}$$

In the following $T - 1$ dialogues, we will specify all user simulators to train the system agent $S$ with this distribution $\boldsymbol{p}$ (lines 15-18). We will also evaluate the RL model $S$ for every $e$ episodes (line 10-12) and update the distribution $\boldsymbol{p}$ with the new $K$ success rates (line 13).

**Difference with the original UCB1.** The main differences between our modified UCB1 algorithm and the original UCB1 algorithm are twofold. First, we tailor the original UCB1 into our scenario by using Eq. 3 to penalize the user simulators with which the dialogue system has performed well. Secondly, we adopt a sampling schema based on a well-designed distribution (see Eq. 4) instead of taking the arm with the highest expectation. This is to increase the diversity and flexibility of arm selection. **We also discuss the differences between our MUST$_\text{adaptive}$ and some other related works in App. H.**

## 5   EXPERIMENTS

To verify the effectiveness of our proposed MUST, we benchmark the system agents trained either with a single user simulator or multiple user simulators (including MUST$_\text{merging}$, MUST$_\text{uniform}$, MUST$_\text{adaptive}$). **For the MUST$_\text{CRL}$ strategy, we will discuss it in the App. C.**

### 5.1   EXPERIMENTAL SETUP

**Available user simulators.** There are six user simulators provided by Shi et al. (2019), which are Agenda-Template (**AgenT**), Agenda-Retrieval (**AgenR**), Agenda-Generation (**AgenG**), RNN-Template (**RNNT**), RNN-Retrieval (**RNNR**), RNN-End2End (**RNN**) trained with different dialog planning and generation methods. The NLU modules of all six user simulators are using the RNN model. The DM modules of **AgenT**, **AgenR**, and **AgenG** are rule-based methods. For the NLG module, these three simulators are using the template, retrieval, and generation methods respectively.

The DM modules of **RNNT**, **RNNR** are using Sequicity (Lei et al., 2018) as their backbones which is an RNN-based seq2seq model with copy mechanism. The NLG modules of these two simulators are using the template and retrieval methods respectively. The user simulator of **RNN** uses Sequicity as its backbone in an end-to-end manner.

**Baselines.** The baselines are the dialogue systems trained by each user simulator, including **Sys-AgenT**, **Sys-AgenR**, **Sys-AgenG**, **Sys-RNNT**, **Sys-RNNR**, and **Sys-RNN**. For a fair comparison, all system agents (including the systems trained by our MUST) have the same architecture described in Shi et al. (2019). See basic modules of user simulators and dialogue systems in App. B.1.

**MultiWOZ Restaurant Domain Dataset.** The original task in MultiWOZ (Budzianowski et al., 2018) is to model the system response. Shi et al. (2019) annotate the user intents and the user-side dialog acts in the restaurant domain of MultiWOZ to build user simulators, which has a total of 1,310 dialogues. Moreover, we randomly simulated 2,000 dialogues from each rule-based simulator (i.e., AgenT, AgenR, AgenG) and their corresponding system agents respectively, and processed these dialogues to have the same annotation format as the MultiWOZ restaurant domain dataset. We denoted this dataset as **Simulated Agenda Dataset**, which has a total of 6,000 dialogues.

**Evaluation Measures.** The direct automatic metric to evaluate the dialogue system is **the success rate** tested by each user simulator. We calculate the success rate between a user simulator and a system agent by sampling 200 dialogues. We exclude some user simulators in MUST training and test the systems with them as **out-of-domain evaluation**. According to the previous study Gunasekara et al. (2020), there usually is a gap between automatic evaluations and human evaluations of dialogue systems. Therefore, we ask human users to converse with dialogue systems. Each dialogue system has conversed with 5 different users; each user has 10 dialogues. In total, we collect 50 dialogues for each dialogue system to calculate its success rate. See more details in App. B.5.

## 5.2 IMPLEMENTATIONS

### 5.2.1 TWO NEW USER SIMULATORS

We believe Pre-trained Language Models (PLMs) might improve the capacity of user simulators since they have recently shown remarkable success in building task-oriented dialogue systems (Ham et al., 2020; Peng et al., 2021; Hosseini-Asl et al., 2020). Here we implement another two user simulators using GPT (Radford et al., 2018; 2019). Building a user simulator using GPT is similar to building a ToD system with GPT. See more details in App. G.

**GPT Simulator.** It is first fine-tuned on the *simulated agenda dataset* and then fine-tuned on the *MultiWOZ restaurant domain dataset* by leveraging GPT. This user simulator will be used to help implementing MUST.

**GPT$_{IL}$ Simulator.** For implementing the MUST$_{merging}$ strategy, similar to Imitation Learning (IL), we first train a new user simulator with dialogue sessions collected from different user simulators and their corresponding dialogue systems. We also learn this new user simulator based on GPT model and denote it as GPT$_{IL}$. GPT$_{IL}$ is first fine-tuned on the *simulated agenda dataset*. Then we sample 1,400 dialogues from the *simulated agenda dataset* and merge them with 1,310 *MultiWOZ restaurant domain dialogues* to continue fine-tuning GPT$_{IL}$.

### 5.2.2 DIALOGUE SYSTEMS

**Sys-GPT** is trained with the *single* user simulator GPT. **Sys-MUST$_{merging}$** is trained with **GPT$_{IL}$**. **Sys-MUST$_{uniform}$** is trained by the user simulators of AgenT, AgenR, RNNT, and GPT with a uniform sampling distribution. For training **Sys-MUST$_{adaptive}$** [6], the distribution $p$ will be adaptively updated using our modified UCB1 algorithm. **We also train the Sys-MUST$_{uniform}$ and Sys-MUST$_{adaptive}$ by using different subsets of the user simulators for ablation studies. See more details in App. D.**

---

[6]See implementations of dialogue systems in App. B.2 and policy gradient algorithm in App. B.3.

Table 2: The success rates of the system agents were tested against various user simulators. Each column represents a user simulator, each row represents a dialogue system trained with a specific simulator, e.g., Sys-AgenT means the system trained with AgenT. Each entry shows the success rate on 200 dialogues collected from a user simulator and a system agent. We use four user simulators: AgenT, AgenR, RNNT, and GPT simulator to implement $\text{MUST}_{\text{uniform}}$ and $\text{MUST}_{\text{adaptive}}$.

| Dialogue Systems | | In-domain evaluation | | | | Out-of-domain evaluation | | | | | All | |
|---|---|---|---|---|---|---|---|---|---|---|---|---|
| | | AgenT | AgenR | RNNT | GPT | AgenG | RNNR | RNN | Avg.↑ | Std.↓ | Avg.↑ | Std.↓ |
| single | Sys-AgenT | 97.5 | 54.0 $\downarrow_{40.0\%}$ | 98.5 $\downarrow_{0.5\%}$ | 78.0 $\downarrow_{4.9\%}$ | 72.5 | 92.5 | 77.0 | 80.7 | 8.6 | 81.4 | 14.8 |
| | Sys-AgenR | 96.0 $\downarrow_{1.5\%}$ | 90.0 | 98.5 $\downarrow_{0.5\%}$ | 80.5 $\downarrow_{1.8\%}$ | 97.5 | 97.5 | 82.0 | 92.3 | 7.3 | 91.7 | 7.1 |
| | Sys-RNNT | 30.5 $\downarrow_{68.7\%}$ | 23.0 $\downarrow_{74.4\%}$ | 99.0 | 75.5 $\downarrow_{7.9\%}$ | 35.5 | 97.5 | 84.0 | 72.3 | 26.6 | 63.6 | 30.5 |
| | **Sys-GPT** | 60.5 $\downarrow_{37.9\%}$ | 51.5 $\downarrow_{42.8\%}$ | 97.0 $\downarrow_{2.0\%}$ | 82.0 | 59.5 | 94.0 | 92.0 | 81.8 | 15.8 | 76.6 | 17.6 |
| MUST | **Sys-MUST**$_{\text{merging}}$ | 97.5 $\uparrow_{0.0\%}$ | 83.5 $\downarrow_{7.2\%}$ | 94.5 $\downarrow_{4.6\%}$ | 80.5 $\downarrow_{1.8\%}$ | 97.5 | 94.0 | 82.5 | 91.3 | 6.4 | 90.0 | 6.9 |
| | **Sys-MUST**$_{\text{uniform}}$ | 97.5 $\uparrow_{0.0\%}$ | 89.0 $\downarrow_{1.0\%}$ | 97.5 $\uparrow_{1.5\%}$ | 82.5 $\uparrow_{0.5\%}$ | 96.5 | 96.0 | 87.5 | 93.4 | 4.2 | 92.4 | 5.6 |
| | **Sys-MUST**$_{\text{adaptive}}$ | 97.5 $\uparrow_{0.0\%}$ | 89.5 $\downarrow_{0.5\%}$ | 97.0 $\downarrow_{2.0\%}$ | 82.5 $\uparrow_{0.5\%}$ | 96.5 | 97.5 | 90.0 | **94.7** | **3.3** | **92.9** | **5.3** |

[1] The underlined number represents the success rate between a user simulator and its corresponding dialogue system trained by this user simulator. The increasing and decreasing percentages (in red and green colors) use the underlined numbers as the base success rates.

[2] ↓ (↑) indicates by what percentages the success rate has decreased (increased) compared with the base success rate by interacting with the same user simulator.

## 5.3 Experimental Results

**Automatic Evaluation.** As seen in Tab. 2, $\text{Sys-MUST}_{\text{uniform}}$ and $\text{Sys-MUST}_{\text{adaptive}}$ outperform the dialogue systems (Sys-AgenT, Sys-AgenR, Sys-RNNT, and Sys-GPT) trained by a single user simulator in the overall performance, demonstrating the superiority of leveraging multiple user simulators. Especially, **Sys-MUST**$_{\text{adaptive}}$ has a 1.2 absolute value improvement (92.9 vs. 91.7) **averagely** over the previous SOTA system Sys-AgenR. Observing that $\text{Sys-MUST}_{\text{merging}}$ is not as competitive as $\text{Sys-MUST}_{\text{uniform}}$ and $\text{Sys-MUST}_{\text{adaptive}}$, this comparison **shows** that the merging strategy cannot effectively leverage multiple user simulators.

In **in-domain evaluation**, the performances of systems (Sys-AgenT, Sys-AgenR, Sys-RNNT, and Sys-GPT) trained by a single user simulator drop a lot when testing with a different user simulator. It requires us to delicately select a suitable user simulator for obtaining a good dialogue system. However, human users might be multi-facet or even unknown, which makes the selection become difficult. Therefore, it is essential to leverage multiple user simulators when training dialogue systems. At least, the performance gap of dialogue systems trained with our MUST becomes smaller than without MUST, see the percentages labeled in green and red colors.

In **out-of-domain evaluation** where the user simulators used for testing the systems are unseen by our MUST, **Sys-MUST**$_{\text{uniform}}$ **and Sys-MUST**$_{\text{adaptive}}$ **achieve at most 2.4 absolute value improvement over Sys-AgenR**. This evidences that MUST has a better generalization ability for interacting with unseen user simulators. Moreover, the dialogue systems ($\text{Sys-MUST}_{\text{merging}}$, $\text{Sys-MUST}_{\text{uniform}}$, and $\text{Sys-MUST}_{\text{adaptive}}$) trained with the proposed MUST approaches have lower standard deviations, which indicates that they are more robust to the diversity of user simulators.

**Human Evaluation.** The human evaluation results in Tab. 3 show that our $\text{Sys-MUST}_{\text{uniform}}$ and $\text{Sys-MUST}_{\text{adaptive}}$ largely outperform the other dialogue systems when interacting with real users. The consistency between automatic evaluations and human evaluations evidences the effectiveness of our proposed MUST.

Table 3: Human evaluation.

| Dialogue Systems | | human evaluation |
|---|---|---|
| single | Sys-AgenT | 76.0 |
| | Sys-AgenR | 84.0 |
| | Sys-RNNT | 34.0 |
| | **Sys-GPT** | 58.0 |
| MUST | **Sys-MUST**$_{\text{merging}}$ | 90.0 |
| | **Sys-MUST**$_{\text{uniform}}$ | 92.0 |
| | **Sys-MUST**$_{\text{adaptive}}$ | 92.0 |

## 5.4 Analysis and discussions

**Convergences between MUST**$_{\text{uniform}}$ **and MUST**$_{\text{adaptive}}$**.** In Fig. 2, we show the learning curves of $\text{Sys-MUST}_{\text{uniform}}$ and $\text{Sys-MUST}_{\text{adaptive}}$ in 100,000 steps; the first 40,000 steps are in the **warm-up phase** for $\text{Sys-MUST}_{\text{adaptive}}$. From Fig. 2(a), we can see that training the dialogue system with AgenT, AgenR, RNNT, and GPT by $\text{MUST}_{\text{adaptive}}$ can converge faster than by $\text{MUST}_{\text{uniform}}$. **We do ablation studies on our *modified* UCB1 algorithm to help understanding the designed distribution $p$, see details in App. E.** We further plot the performances of the dialogue system tested by each user simulator in the RL training, which is shown in Fig. 2(b)-2(e).

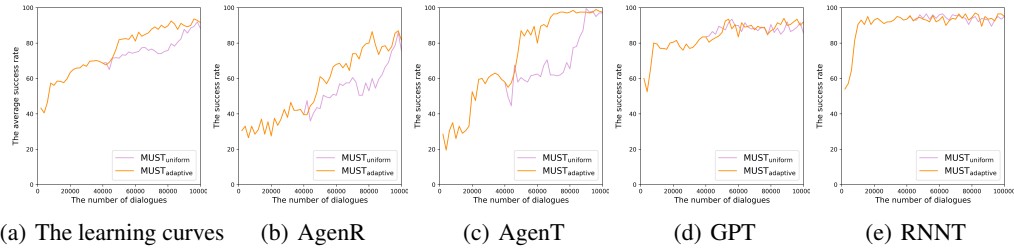

| (a) The learning curves | (b) AgenR | (c) AgenT | (d) GPT | (e) RNNT |
|---|---|---|---|---|

Figure 2: The learning curves of Sys-MUST$_{\text{uniform}}$ and Sys-MUST$_{\text{adaptive}}$. (a) shows their average success rates tested with all user simulators (AgenT, AgenR, RNNT, and GPT). The success rates of them tested with each user simulator are shown in (b)-(e).

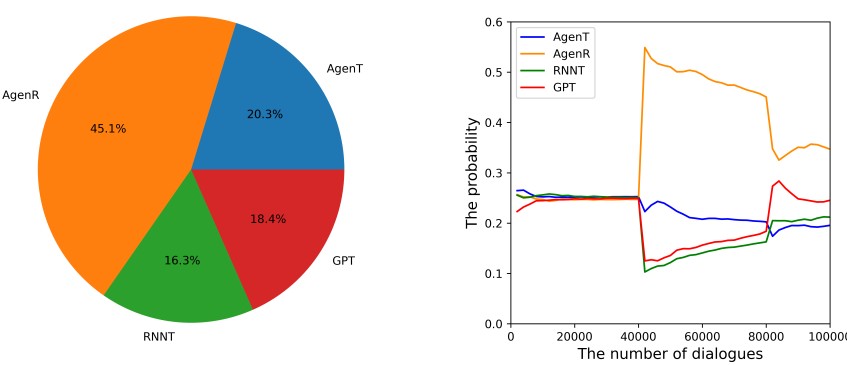

(a) The sampling proportion of simulators.

(b) Variations of the sampling proportions (in every 2000 steps) of simulators.

Figure 3: The sampling proportions of user simulators in average (a) and in time horizon (b).

**Visualization of the patterns learned by MUST$_{\text{adaptive}}$.** Let us define the adaptation difficulty of a user simulator using how many steps it must take to train the dialogue system with this user simulator until it converges. The adaptation difficulty of all user simulators could be ranked like AgenR > AgenT > GPT > RNNT according to Fig. 2(b)-2(e). To check whether MUST$_{\text{adaptive}}$ tends to sample harder-to-adapt user simulators more times in the **adaptive phase**, as assumed in Sec. 4.2, we visualize the sampling proportions of all user simulators in Fig. 3(a). We could observe that AgenR was sampled with 45.1% (the biggest proportion) and it is indeed the hardest user simulator that can be adapted by the system; RNNT has the smallest sampling proportion and it is the easiest user simulator that can be adapted by the system. The consistency between the adaptation difficulty and sampling proportions for these four user simulators evidences our assumption in Sec. 4.2. Fig. 3(b) visualizes the variations of the sampling distributions of user simulators. Interestingly, it shows that AgenR and AgenT are *competitive* with the GPT simulator; while RNNT and GPT are *cooperative* with each other. This might be because RNNT and GPT simulator are learned from the dialogue corpus and will share similar behaviors.

## 6 CONCLUSION

In this paper, we propose a framework named MUST to improve the system agent by using multiple user simulators simultaneously. We discuss several simple methods to implement MUST, which is either inflexible or inefficient. Therefore, we formulate MUST as a Multi-armed bandits (MAB) problem, based on which we propose a novel implementation called MUST$_{\text{adaptive}}$. The experimental results on the restaurant search task from MultiWOZ demonstrate that our proposed MUST can largely improve the system agent upon the baseline methods, especially when tested with unseen user simulators. Moreover, MUST$_{\text{adaptive}}$ is robust to the diversity of user simulators and its training is more efficient than MUST$_{\text{uniform}}$. The main limitation of this work is that we only conduct our experiments on the restaurant domain of the MultiWOZ since we can only find multiple user simulators from Shi et al. (2019) and they build these simulators only on the restaurant search task. In future work, we plan to apply our proposed methods to multi-domain scenarios.

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

## A    MULTI-ARMED BANDIT PROBLEM

Reinforcement learning policies face the exploitation versus exploration trade-off, which can be described as the search for a balance between exploring the environment to find profitable actions while taking the empirically best action as often as possible. This exploitation vs exploration dilemma has been widely studied as a Multi-armed bandit (MAB) problem.

In the MAB problem, there are $K$ arms, and each arm $j$ has a fixed but unknown reward distribution $R_j$ with an expectation $\mu_j$. At each time step $t = 1, 2, ..., T$, the decision maker must choose one of these $K$ arms. We denote the arm pulled at time step $t$ as $j_t \in \{1, ..., K\}$. After pulling an arm, it will receive a reward $X_{j_t}$ which is a realization drawn from the arm's underlying reward distribution. The decision masker's objective is to maximize the cumulative expected reward over the time horizon $\sum_{t=1}^{T} \mathbb{E}[X_{j_t}] = \sum_{t=1}^{T} \mu_{j_t}$.

## B    MORE DETAILS ABOUT TRAINING DIALOGUE SYSTEMS

### B.1    THE ARCHITECTURES OF USER SIMULATORS AND DIALOGUE SYSTEMS

The basic modules of user simulators and dialogue systems are detailed in Tab. 4

Table 4: The architectures of user simulators and dialogue systems. Modules with $^{\dagger}$ are trainable.

| Agent Types | Agents | NLU | DM | NLG |
|---|---|---|---|---|
| User Simulators | **AgenT** (Shi et al., 2019) | **RNN**$^{\dagger}$ | Agenda | Template |
| | **AgenR** (Shi et al., 2019) | **RNN**$^{\dagger}$ | Agenda | Retrieval |
| | **AgenG** (Shi et al., 2019) | **RNN**$^{\dagger}$ | Agenda | **RNN**$^{\dagger}$ (Generation) |
| | **RNNT** (Shi et al., 2019) | **RNN**$^{\dagger}$ | | Template |
| | **RNNR** (Shi et al., 2019) | **RNN**$^{\dagger}$ | | Retrieval |
| | **RNN** (Shi et al., 2019) | **RNN**$^{\dagger}$ (NLU + NLG) | | |
| | GPT (ours) | **Transformer**$^{\dagger}$ (NLU + DM + NLG) | | |
| | GPT$_{\text{IL}}$ (ours) | **Transformer**$^{\dagger}$ (NLU + DM + NLG) | | |
| Dialogue Systems | All | **RNN**$^{\dagger}$ | **RNN**$^{\dagger}$ | Template |

### B.2    THE IMPLEMENTATIONS OF THE DIALOGUE SYSTEMS

The NLU modules of all system agents are a 2-layer bidirectional-GRU with 200 hidden units. The NLG modules of them are using the template-based method. The DM modules of them are a simple MLP. The input of the DM module is a state representation, which consists of the traditional dialog state and word count vector of the current utterance same as Shi et al. (2019). We mainly use the policy gradient method to train the DM modules of dialogue systems from scratch.

### B.3    THE DETAILS OF RUNNING POLICY GRADIENT ALGORITHM

For training the DM modules of dialogue systems with the policy gradient method, we also apply the $\epsilon$-greedy exploration strategy. We let $\epsilon$ be 0.5 in the beginning, and it will decrease to 0 linearly within the RL training. The dialogue ends either when the user simulators say "goodbye" or when the number of turns of the dialogue exceeds 10. The reward will be given +1 for task success, -1 for task failure, and -0.1 for each additional turn to encourage the RL-based policy module to finish the task fast. Also, a discounted factor of 0.9 is applied to all the experiences.

### B.4    THE PARAMETERS OF TRAINING SYS-MUST$_{\text{adaptive}}$

The hyperparameters used to train the Sys-MUST$_{\text{adaptive}}$ are listed in the Tab. 5.

Table 5: The hyperparameters used for training the Sys-MUST$_{\text{adaptive}}$.

| Hyperparameter | Value |
|:---:|:---:|
| $T$ | 100,000 |
| $T_0$ | 40,000 |
| $e$ | 2,000 |
| $d$ | 200 |
| $\tau$ | 0.75 |

Table 6: The relationships between user acts and system acts.

| User act | System act |
|---|---|
| inform type | ask type, present result, nomatch result |
| inform type change | ask type, present result, nomatch result |
| anything else | present result, no other |
| make reservation | ask reservation info, booking success, booking fail |
| make reservation change time | ask reservation info, booking success, booking fail |
| ask info | provide info |
| goodbye | goodbye |

### B.5 HUMAN EVALUATION ON DIALOGUE SYSTEMS

We find 5 volunteers to conduct the human evaluations on dialogue systems. They all have good English skills and are unpaid. Before the experiments, we introduced task-oriented dialogue systems and user simulators to them and tell them how to judge if the generated dialogue is successful. Then we prepare 50 user goals from **MultiWOZ Restaurant Domain Dataset**: 20 of them are simple, and 30 of them are a little bit complex. We specify 10 user goals for each volunteer and let the volunteer converse with all dialogue systems for each same user goal. In total, we collect 50 dialogues for each dialogue system to calculate its success rate.

**The criteria to judge whether a task-oriented dialogue is successful are based on two aspects: 1) the system agent correctly understands the user's goal (i.e., the predicted dialogue state tracking result is correct); and 2) the system agent provides all information (i.e., all slot values or a booking reference number) that the user requests. For human evaluations, we follow these standard criteria. Besides, we also see if the system act generated by the system agent is matched to the user act for each turn in the dialogue.**

**There have seven user acts, which are 'inform type", "inform type change", "ask info", "anything else", "make reservation", "make reservation change time", and "goodbye". There have nine system acts, which are "ask type", "present result", "nomatch result", "no other", "ask reservation info", "provide info", "booking success", "booking fail" and "goodbye". The relationships between user acts and system acts are shown in Tab. 6.**

### C IMPLEMENT MUST WITH THE MUST$_{\text{CRL}}$ STRATEGY

Without losing any generality, we consider two representative sequential orders: 1) AgenT, AgenR, RNNT, GPT; and 2) AgenR, GPT, AgenT, RNNT. For case 1, the first two user simulators are Agenda-based user simulators; the last two user simulators are Neural networks-based user simulators. For case 2, we interleave these two types of user simulators. When the system trained by a user simulator converges, we let it continue to interact with another user simulator following the order.

As seen in Tab. 7, in case 1, the system agent achieves the best performance (i.e., 92.4 in terms of the average success rate) after training with AgenT and AgenR sequentially. However, its overall performance degrades to 83.0 after training with RNNT; especially, its performance decreases by 36.0% when testing with AgenR ($93.0 \rightarrow 59.5$). Moreover, after continuing to learn from GPT, the performance of the system agent becomes worse for AgenT ($95.0 \rightarrow 75.5$) and AgenR ($59.5 \rightarrow 47.5$). This indicates the catastrophic forgetting issue heavily happened when the system agent starts learning from AgenR. We also could observe a similar phenomenon from case 2. These results can

Table 7: The experimental results of implementing MUST with the $MUST_{CRL}$ strategy.

| | Dialogue Systems | User simulators | | | | |
|---|---|---|---|---|---|---|
| | | AgenT | AgenR | RNNT | GPT | Avg. |
| **Case 1** | trained by *AgenT* | 97.5 | 54.0 | 98.5 | 78.0 | 82.0 |
| | trained by *AgenT, AgenR* sequentially | $97.0_{\downarrow 0.5\%}$ | 93.0 | 97.0 | 82.5 | **92.4** |
| | trained by *AgenT, AgenR, RNNT* sequentially | $95.0_{\downarrow 2.6\%}$ | $59.5_{\downarrow 36.0\%}$ | 97.0 | 80.5 | 83.0 |
| | trained by *AgenT, AgenR, RNNT, GPT* sequentially | $75.5_{\downarrow 22.6\%}$ | $47.5_{\downarrow 48.9\%}$ | $96.0_{\downarrow 1.0\%}$ | 82.0 | 75.3 |
| **Case 2** | trained by *AgenR* | 96.0 | 90.0 | 98.5 | 82.5 | **91.8** |
| | trained by *AgenR, GPT* sequentially | 97.5 | $88.0_{\downarrow 2.2\%}$ | 97.0 | 81.5 | 91.0 |
| | trained by *AgenR, GPT, AgenT* sequentially | 96.5 | $78.5_{\downarrow 12.8\%}$ | 97.0 | $80.0_{\downarrow 1.8\%}$ | 88.0 |
| | trained by *AgenR, GPT, AgenT, RNNT* sequentially | $97.5_{\uparrow 1.0\%}$ | $65.5_{\downarrow 27.2\%}$ | 95.0 | $78.5_{\downarrow 3.7\%}$ | 84.1 |

Table 8: Ablation study on MUST. It uses **five user simulators (AgenT, AgenR, RNNT, RNNR and GPT simulator)** to implement $MUST_{uniform}$ and $MUST_{adaptive}$.

| Dialogue Systems | | In-domain evaluation | | | | | Out-of-domain evaluation | | | | All | |
|---|---|---|---|---|---|---|---|---|---|---|---|---|
| | | AgenT | AgenR | RNNT | RNNR | GPT | AgenG | RNN | Avg.↑ | Std.↓ | Avg.↑ | Std.↓ |
| **single** | Sys-AgenT | 97.5 | $54.0_{\downarrow 40.0\%}$ | $98.5_{\downarrow 0.5\%}$ | $92.5_{\downarrow 1.0\%}$ | $78.0_{\downarrow 4.9\%}$ | 72.5 | 77.0 | 74.8 | **2.3** | 81.4 | 14.8 |
| | Sys-AgenR | $96.0_{\downarrow 1.5\%}$ | 90.0 | $98.5_{\downarrow 0.5\%}$ | $97.5_{\uparrow 4.3\%}$ | $80.5_{\downarrow 1.8\%}$ | 97.5 | 82.0 | 89.8 | 7.8 | 91.7 | 7.1 |
| | Sys-RNNT | $30.5_{\downarrow 68.7\%}$ | $23.0_{\downarrow 74.4\%}$ | 99.0 | $97.5_{\uparrow 4.3\%}$ | $75.5_{\downarrow 7.9\%}$ | 35.5 | 84.0 | 59.8 | 24.3 | 63.6 | 30.5 |
| | Sys-RNNR | $30.0_{\downarrow 68.7\%}$ | $23.0_{\downarrow 74.4\%}$ | $96.5_{\downarrow 2.5\%}$ | 93.5 | $68.5_{\downarrow 16.5\%}$ | 30.0 | 70.5 | 50.3 | 20.3 | 58.9 | 28.8 |
| | **Sys-GPT** | $60.5_{\downarrow 37.9\%}$ | $51.5_{\downarrow 42.8\%}$ | $97.0_{\downarrow 2.0\%}$ | $94.0_{\uparrow 0.5\%}$ | 82.0 | 59.5 | 92.0 | 75.8 | 16.3 | 76.6 | 17.6 |
| | **Sys-MUST**$_{uniform}$ | $97.5_{\uparrow 0.0\%}$ | $87.0_{\downarrow 3.3\%}$ | $97.0_{\downarrow 2.0\%}$ | $97.5_{\uparrow 4.3\%}$ | $82.0_{\uparrow 0.0\%}$ | 96.5 | 87.0 | 91.8 | 4.8 | 92.1 | 6.0 |
| | **Sys-MUST**$_{adaptive}$ | $97.0_{\downarrow 0.5\%}$ | $89.0_{\downarrow 1.1\%}$ | $97.0_{\downarrow 2.0\%}$ | $97.5_{\uparrow 4.3\%}$ | $82.5_{\uparrow 0.6\%}$ | 97.5 | 87.5 | **92.5** | 5.0 | **92.6** | **5.7** |

confirm that implementing our proposed MUST with $MUST_{CRL}$ strategy indeed has the catastrophic forgetting issue.

# D   SENSITIVITY ON DIFFERENT SUBSETS OF USER SIMULATORS

We also train the Sys-MUST$_{uniform}$ and Sys-MUST$_{adaptive}$ by using different groups of user simulators for ablation studies: 1) five user simulators of AgenT, AgenR, RNNT, RNNR, and GPT; and 2) three user simulators including AgenT, RNNT, and GPT.

**Superiority of MUST.**   From Tab. 8 and Tab. 9, we can observe that Sys-MUST$_{uniform}$ and Sys-MUST$_{adaptive}$ largely outperform the dialogue systems trained by single user simulators. Especially, they gain an improvement of 4 absolute points (85.4 vs. 81.4) when trained with three user simulators of AgenT, RNNT, and GPT. In summary, MUST could consistently improve the performance of the systems when using different numbers of user simulators. The ablation studies on different subsets of user simulators can demonstrate the robustness of MUST.

**Out-of-domain evaluation.**   When testing our MUST with unseen user simulators, Sys-MUST$_{uniform}$ and Sys-MUST$_{adaptive}$ can also largely outperform the dialogue systems trained by a single user simulator. As seen in Tab. 8, Sys-MUST$_{adaptive}$ achieves a 2.7 absolute value improvement (92.5 vs 89.8) over Sys-AgenR. Sys-MUST$_{uniform}$ and Sys-MUST$_{adaptive}$ even improve at least 5.7 points (80.0 vs 74.3) over Sys-GPT (as shown in Tab. 9). These experimental results on different subsets of user simulators demonstrate that our MUST has a better generalization ability for interacting with unseen user simulators and is insensitive to the user simulator selection.

**Comparison between MUST$_{uniform}$ and MUST$_{adaptive}$.**   Fig. 4 shows the learning curves of Sys-MUST$_{uniform}$ and Sys-MUST$_{adaptive}$ on different subsets of user simulators. The first 40,000 steps are in the **warm-up phase** for Sys-MUST$_{adaptive}$. We could conclude that training the dialogue system by MUST$_{adaptive}$ consistently converges faster than by MUST$_{uniform}$, at least in the scenarios when using three, four, or five user simulators to implement MUST (see Fig. 4(a), Fig. 2(a), and Fig. 4(b), respectively).

From Tab. 8 where MUST is trained with five user simulators, we could observe that Sys-MUST$_{adaptive}$ outperforms Sys-MUST$_{uniform}$ with 0.5 absolute point. The performance gain becomes smaller when MUST is trained with three user simulators (see Tab. 9). This probably shows that Sys-MUST$_{adaptive}$ would be more beneficial when there exist more user simulators.

Table 9: Ablation study on MUST. It uses **three user simulators (AgenT, RNNT, and GPT simulator)** to implement MUST$_{uniform}$ and MUST$_{adaptive}$.

| Dialogue Systems | | In-domain evaluation | | | Out-of-domain evaluation | | | | | | All | |
|---|---|---|---|---|---|---|---|---|---|---|---|---|
| | | AgenT | RNNT | GPT | AgenR | AgenG | RNNR | RNN | Avg.↑ | Std.↓ | Avg.↑ | Std.↓ |
| single | Sys-AgenT | 97.5 | 98.5$_{\downarrow 0.5\%}$ | 78.0$_{\downarrow 0.5\%}$ | 54.0 | 72.5 | 92.5 | 77.0 | 74.0 | 13.7 | 81.4 | 14.8 |
| | Sys-RNNT | 30.5$_{\downarrow 68.7\%}$ | 99.0 | 75.5$_{\downarrow 7.9\%}$ | 23.0 | 35.5 | 97.5 | 84.0 | 60.0 | 31.4 | 63.6 | 30.5 |
| | Sys-GPT | 60.5$_{\downarrow 37.9\%}$ | 97.0$_{\downarrow 2.0\%}$ | 82.0 | 51.5 | 59.5 | 94.0 | 92.0 | 74.3 | 19.0 | 76.6 | 17.6 |
| MUST | **Sys-MUST$_{uniform}$** | 97.5$_{\uparrow 0.0\%}$ | 96.0$_{\downarrow 3.0\%}$ | 82.5$_{\uparrow 0.6\%}$ | 55.0 | 82.0 | 97.5 | 87.0 | 80.3 | 15.7 | **85.4** | **13.9** |
| | **Sys-MUST$_{adaptive}$** | 97.5$_{\uparrow 0.0\%}$ | 97.5$_{\downarrow 1.5\%}$ | 82.5$_{\uparrow 0.6\%}$ | 55.5 | 80.5 | 97.0 | 87.0 | 80.0 | 15.3 | **85.4** | **13.9** |

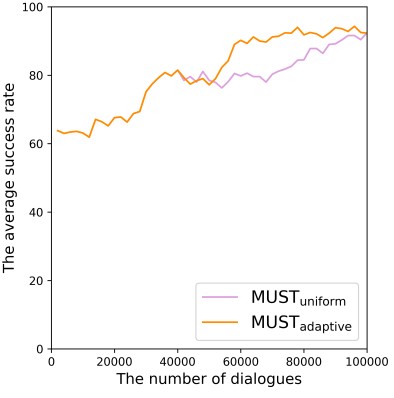
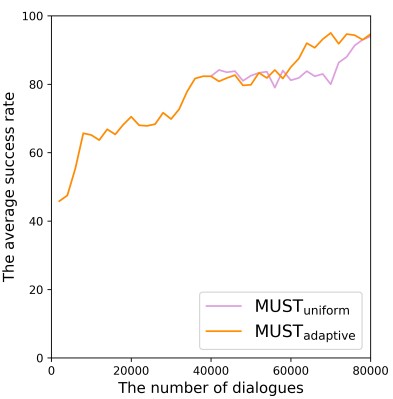

(a) MUST with five use simulators      (b) MUST with three use simulators

Figure 4: The learning curves of Sys-MUST$_{uniform}$ and Sys-MUST$_{adaptive}$.

# E  ABLATION STUDY FOR THE MODIFIED UCB1 ALGORITHM

## E.1  NECESSITY OF THE EXPLORATION TERM

Our *modified* UCB1 algorithm provides a distribution for guiding how to sample different user simulators to accelerate the entire MUST training. The exploration term in the proposed MUST$_{adaptive}$ exists mainly for uniform adaption (see the detailed explanation in Sec. 4.1). The original UCB1 algorithm (Auer et al., 2002) can tell us how to pull arms in bandits to maximize the cumulative expected reward. It is well-known that it cannot explore effectively without the exploration (UCB) term; consequently, it might not find the optimal action and lead to relatively poor performance. It is difficult to theoretically prove the usefulness of the exploration term in our scenario (like in the original UCB1 algorithm), which we leave as future work. However, we alternatively conduct some ablation studies to evidence the necessity of the exploration term.

**MUST$_{adaptive}$w/t exploration**    If we omit the exploration term in our *modified* UCB1 algorithm, the simplest way to calculate the distribution $p$ is to make the sample probability w.r.t a user simulator solely depend on the inversion of the system's performance. See the row called 'w/t exploration' in Tab. 10 for comparisons.

In this situation, the obtained distribution $p$ might be sharp due to the lack of the exploration term, which would be harmful for uniform adaption to some extent. As Fig. 5(a) shows, MUST$_{adaptive}$ performs worse and converges slower when omitting the exploration term, compared with when our *modified* UCB1 algorithm has the exploration term. This could demonstrate both the importance of uniform adaption and the usefulness of the exploration term.

## E.2  ABLATION STUDY ON THE DESIGNED DISTRIBUTION

**Rationale of exploitation vs exploration trade-off.**    Similar to the exploitation vs exploration trade-off, the distribution $p$ under the MUST$_{adaptive}$ should trade off the boosting adaption and the uniform adaption when specifying multiple user simulators. Considering the boosting adaption, we make a *exploitation assumption* stated as follows: $p$ is expected to *assign lower weights to*

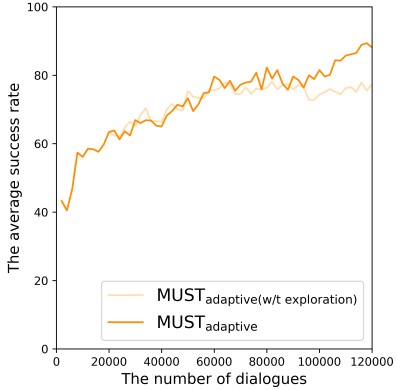 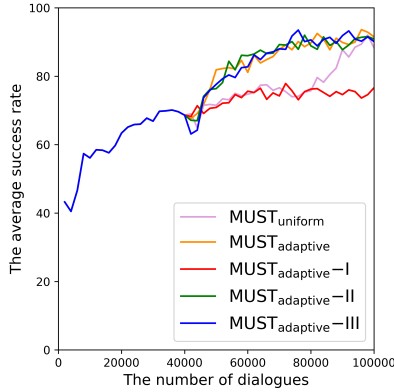

(a) Ablation study on the exploration term  (b) Ablation study on the distribution $\boldsymbol{p}$

Figure 5: The learning curves of Sys-MUST$_{\text{uniform}}$ and Sys-MUST$_{\text{adaptive}}$.

*user simulators with which the system agent $S$ already performs well* and *higher weights to those user simulators with which $S$ performs poorly*. Therefore, the sampling ratios for different user simulators should be inversely proportional to the system's performance on each user simulator.

**Rationale of the modified UCB1 algorithm**   The modified UCB1 algorithm for implementing MUST$_{\text{adaptive}}$ is defined as

$$
\hat{x}_j = \underbrace{\bar{x}_j}_{\text{exploitation}} + \underbrace{\sqrt{\frac{2\ln t}{T_{j,t}}}}_{\text{exploration}}, j \in \{1, ..., K\};
$$
$$
z_j = 1/\left(\hat{x}_j - \tau \min(\{\bar{x}_1, \cdots, \bar{x}_K\})\right),
$$
$$
\boldsymbol{p}_i = \frac{z_j}{\sum_{j=1}^{K} z_j}.
$$

(5)

MUST$_{\text{adaptive}}$ in Eq. 5 (which is the same as Eq. 2, Eq. 3, and Eq. 4) consists of three steps: exploitation-exploration term construction, post-processing (re-scaling operation and the inversion operation), and the probability normalization, corresponding to each line in Eq. 5. Besides this way, we could have the following three variants that shuffle the order of these three key operations (i.e., the exploitation-exploration term construction, re-scaling operation, and the inversion operation). We name these variants as as MUST$_{\text{adaptive}}$-I, MUST$_{\text{adaptive}}$-II, and MUST$_{\text{adaptive}}$-III.

**MUST$_{\text{adaptive}}$-I.**   For the exploitation assumption, we make the exploitation term inversely proportional to the system's performance $\bar{x}_j$ on each user simulator $U_j$, which is denoted as MUST$_{\text{adaptive}}$-I. From Tab. 10, we can obverse that the difference between MUST$_{\text{adaptive}}$-I and MUST$_{\text{adaptive}}$ is that MUST$_{\text{adaptive}}$-I take the inversion of $\bar{x}$ before the exploitation-exploration term construction while MUST$_{\text{adaptive}}$ take the inversion operation after the exploitation-exploration term construction. Since each $\bar{x}_j, j \in \{1, \cdots, K\}$ is smaller than 1, $\frac{1}{\bar{x}_j}$ will be larger than 1. Therefore, the term of $\frac{1}{\bar{x}_j}$ and the exploration term of $\sqrt{\frac{2\ln t}{T_{j,t}}}$ (smaller than 1) are not with the same magnitude, which will lead to a consequence that the exploitation term becomes dominant while the exploration term is negligible. We have discussed a similar issue of ignoring the exploration term in Sec. E.1. Therefore, we adopt MUST$_{\text{adaptive}}$ in default if not specified rather than MUST$_{\text{adaptive}}$-I since the latter might suffer from the different magnitudes of the exploitation term and the exploration term.

**MUST$_{\text{adaptive}}$-II and MUST$_{\text{adaptive}}$-III.**   Compared to MUST$_{\text{adaptive}}$, MUST$_{\text{adaptive}}$-II moves the inversion operation to the front of the constructed exploitation-exploration term. Likewise, MUST$_{\text{adaptive}}$-III moves the re-scaling and the inversion operations to the front of the constructed exploitation-exploration term. MUST$_{\text{adaptive}}$-II and MUST$_{\text{adaptive}}$-III are used to check the order sensitivity about the exploitation-exploration term construction, re-scaling operation, and the inversion of $\bar{x}_j, j \in \{1, \cdots, K\}$.

Table 10: The variants of $\text{MUST}_{\text{adaptive}}$. The $\text{MUST}_{\text{adaptive}}$ implementation is an exploitation-exploration term followed by a post-processing for the re-scaling purpose and a sum-one normalization. Since we omit the exploration term for the second row, therefore, it does not need the post-processing. $\text{MUST}_{\text{adaptive}}$-III moves the re-scaling and the inversion operations to the front of the constructed exploitation-exploration term.

| variants | exploitation-exploration term | post-processing | distribution |
|---|---|---|---|
| $\text{MUST}_{\text{adaptive}}$ | $\hat{x}_j = \bar{x}_j + \sqrt{\frac{2\ln t}{T_{j,t}}}$ | $z_j = \frac{1}{\left(\hat{x}_j - \tau \min(\{\bar{x}_1, \cdots, \bar{x}_K\})\right)}$ | |
| w/t exploration | $z_j = \frac{1}{\bar{x}_j}$ | | |
| $\text{MUST}_{\text{adaptive}}$-I | $\hat{x}_j = \frac{1}{\bar{x}_j} + \sqrt{\frac{2\ln t}{T_{j,t}}}$ | $z_j = \hat{x}_j - \tau \min(\{1/\bar{x}_1, \cdots, 1/\bar{x}_K\})$ | $\boldsymbol{p}_j = \frac{z_j}{\sum_{i=1}^{K} z_i}$ |
| $\text{MUST}_{\text{adaptive}}$-II | $\hat{x}_j = \frac{1/\bar{x}_j}{\sum_{i=1}^{K} 1/\bar{x}_i}$ $\hat{z}_j = \hat{x}_j + \sqrt{\frac{2\ln t}{T_{j,t}}}$ | $z_j = \hat{z}_j - \tau \min(\{\hat{x}_1, \cdots, \hat{x}_K\})$ | |
| $\text{MUST}_{\text{adaptive}}$-III | $\hat{x}_j = \frac{1}{\left(\bar{x}_j - \tau \min(\{\bar{x}_1, \cdots, \bar{x}_K\})\right)}$ $z_j = \frac{\hat{x}_j}{\sum_{i=1}^{K} \hat{x}_i} + \sqrt{\frac{2\ln t}{T_{j,t}}}$ | | |

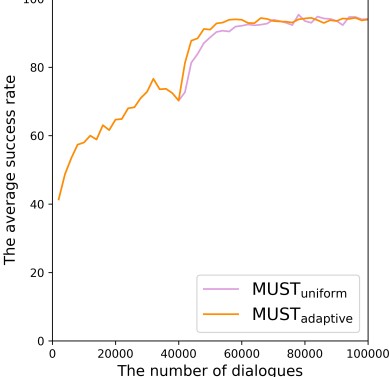

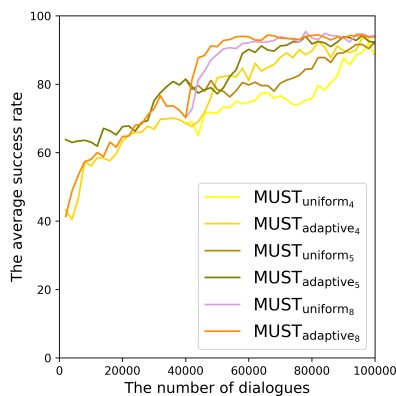

(a) The learning curves of the system trained with eight user simulators.

(b) Comparison between different numbers of user simulators.

Figure 6: The learning curves of Sys-$\text{MUST}_{\text{uniform}}$ and Sys-$\text{MUST}_{\text{adaptive}}$.

**Results for ablation study on the variants.** Experimental results of these different variants are shown in Fig. 5(b). The convergence speed of $\text{MUST}_{\text{adaptive}}$-I is much slower compared to others, which demonstrates that the exploration term is useful once more. The convergence speeds of $\text{MUST}_{\text{adaptive}}$-II and $\text{MUST}_{\text{adaptive}}$-III is comparative to $\text{MUST}_{\text{adaptive}}$. This probably shows that our design with three operations (i.e., exploitation-exploration term construction, re-scaling strategy, and the inversion of $\bar{x}_j$) is not only reasonable but also robust to the order permutation of these three operations.

# F IMPLEMENTING MUST WITH MORE USER SIMULATORS

To implement our MUST with more user simulators, we use *Simulated Agenda Dataset* to train four extra user simulators [7]. Fig. 6(a) shows the learning curve of the system agent trained by MUST with eight simulators (AgenT, AgenR, RNNT, GPT, $\text{GPT}_{\text{AT}}$, $\text{GPT}_{\text{AR}}$, $\text{GPT}_{\text{AG}}$, and $\text{GPT}_{\text{rand}}$). We could observe that the training of our proposed MUST can still succeed when we increase the number of user simulators to eight. Sys-$\text{MUST}_{\text{adaptive}}$ still converges faster than Sys-$\text{MUST}_{\text{uniform}}$ even

---

[7]Simulated Agenda Dataset (See Sec. 5.1) is simulated from each rule-based user simulator (i.e., AgenT, AgenR, AgenG) and its corresponding system agent respectively. We use them to build three new user simulators denoted as $\text{GPT}_{\text{AT}}$, $\text{GPT}_{\text{AR}}$, and $\text{GPT}_{\text{AG}}$ based on the GPT model respectively. For example, we use the simulated dialogues from AgenT and Sys-AgenT to build the $\text{GPT}_{\text{AT}}$. we also collect 3000 dialogues randomly from Simulated Agenda Dataset to train another new GPT user simulator denoted as $\text{GPT}_{\text{rand}}$.

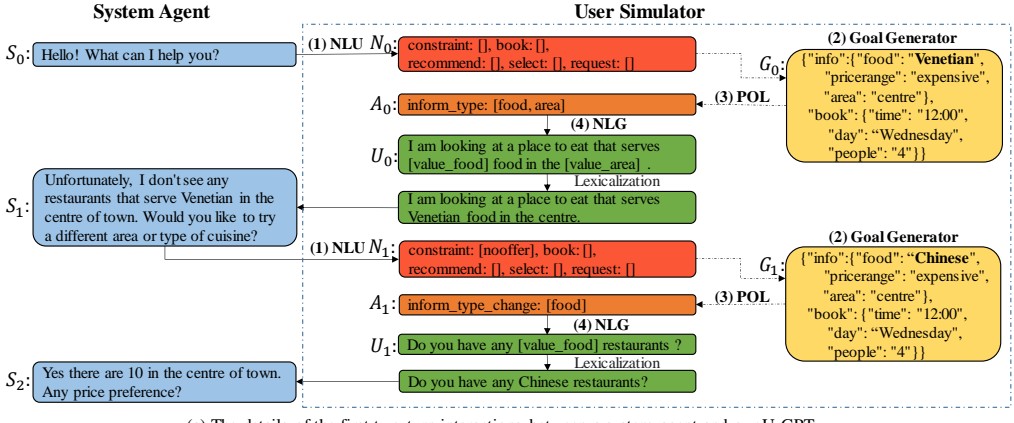

(a) The details of the first two-turn interactions between a system agent and our U-GPT.

hello! what can i help you? <eos_resp> [eos_constraint] [eos_book] [eos_recommend] [eos_select] [eos_request] <eos_nlu> [info] food venetian pricerange expensive area centre [request] [book] time 12:00 day wednesday people 4 <eos_goal> [inform_type] food area <eos_pol> i am looking at a place to eat that serves venetian food in the centre. <eos_utt> unfortunately, i do not see any restaurants that serve venetian in the centre of town. would you like to try a different area or type of cuisine? <eos_resp> nooffer [eos_constraint] [eos_book] [eos_recommend] [eos_select] [eos_request] <eos_nlu> [info] food chinese pricerange expensive area centre [request] [book] time 12:00 day wednesday people 4 <eos_goal> [inform_type_change] food <eos_pol> Do you have any [value_food] restaurants ? <eos_utt>

(b) An example of the model input for training U-GPT.

Figure 7: The overview of our U-GPT which consists of Natural Language Understanding (NLU), Goal Generator, Dialog Policy Learning (POL), and Natural Language Generation (NLG). It uses the auto-regressive language model GPT to understand the system inputs, generate the user actions and the user utterances given the dialogue context and the user goals sequentially in an end-to-end manner. (a) gives a detailed description of the first two-turn interactions between a system agent and our U-GPT. For training U-GPT, we need to convert the dialogue context and all annotations to sequences of tokens. (b) presents the training example of the first two-turn dialogues in (a).

though the difference between their convergence speeds is not too large in this case. It might be because some user simulators are similar (e.g., $GPT_{AT}$ is similar to AgenT, $GPT_{AR}$ is similar to AgenR), which might lead that the distribution $p$ approaches a uniform distribution.

Fig. 6(b) compares the learning curves of Sys-MUST$_{adaptive}$ and Sys-MUST$_{uniform}$ trained with different numbers of user simulators (i.e., four, five, and eight user simulators). It is a fair comparison because these combinations include the hardest user simulator AgenR that can be adapted by the system and the easiest user simulator RNNT that can be adapted by the system (See Sec. 5.4). We can observe that, with more user simulators, Sys-MUST$_{adaptive}$ not only performs better but also converges faster than with fewer user simulators. This probably shows that Sys-MUST$_{adaptive}$ has the potential to be generalized to a larger set of user simulators. Plus, we also could observe that Sys-MUST$_{adaptive}$ consistently converges faster than Sys-MUST$_{uniform}$ in different numbers of user simulators.

## G MODELING USER SIMULATOR WITH GPT

We name the model of building a user simulator based on GPT as U-GPT. In this section, we will illustrate its details and conduct experiments to prove that it is a better model for building a user simulator.

### G.1 THE ARCHITECTURE OF U-GPT

As Fig. 7(a) shown, our U-GPT consists of four modules, which are Natural Language Understanding (NLU), Goal Generator, Dialog Policy Learning (POL), and Natural Language Generation (NLG). Dialogues consist of multiple turns. In the first turn $t = 0$, U-GPT (1) first outputs its NLU results $N_0$ by understanding the system input $S_0$, and (3) decide its actions $A_0$ which is a list of pairs: (action_type, slot_name) based on (2) its initial goal $G_0$ and $\{S_0, N_0\}$. U-GPT then (4) conditions on $\{S_0, N_0, G_0, A_0\}$ to generate the delexicalized utterance $U_0$. The generated placeholders

in $U_0$ will be filled using the corresponding slot values in the goal $G_0$. When the conversation proceeds to turn $t$, U-GPT (1) generates the NLU results $N_t$ based on all of previous dialogue history and generated outputs $\{C_0, \ldots, C_{t-1}, S_t\}$, here $C_i = [S_i, N_i, G_i, A_i, U_i]$. If there has "no-offer" intent in $N_t$ representing that no entities could satisfy current constraints, then (2) Goal Generator should generate a new goal $G_t$. Then U-GPT will continue to (3) generate the user acts $A_t$ and (4) generate delexicalized utterance $U_t$ conditioned on $\{C_0, \ldots, C_{t-1}, S_t, N_t, G_t\}$ sequentially. We should notice that the user utterances occurred in the history context should be lexicalized because they contain important information.

Fig. 7(b) shows an example of training sequence which consists of the concatenation $x = [C_0, C_1]$. In order to leverage GPT, we need to convert the generated outputs $\{N_i, G_i, A_i, U_i\}$ to sequences of tokens resembling a text. And we introduce delimiter tokens *[eos_resp], [eos_nlu], [eos_goal], [eos_pol], [eos_utt]* to signal the ending of sequence representations of different modules. For the NLU results $N_t$, we use five categories: "inform", "request", "book inform", "select", "recommend" same as Shi et al. (2019) to represent them. And we also introduce five tokens *[eos_constraint], [eos_book], [eos_recommend], [eos_select], [eos_request]* to record different information. All of these tokens and the intents of user actions will be added to the vocabulary of GPT as additional special tokens. For training U-GPT, we use the same training objective as GPT which is to maximize the following likelihood:

$$L(U) = \sum_i \log P(u_i | u_{i-k}, ..., u_{i-1}; \Theta),$$

$$\forall\, u_i \in \{S_0, N_0, G_0, A_0, U_0, ..., A_t, U_t\},$$

where $k$ is the size of the context window, and the conditional probability $P$ is parameterized with $\Theta$.

## G.2 EVALUATIONS ON U-GPT

To evaluate our proposed U-GPT, we adopt both **indirect** evaluations and **direct** evaluations as in Shi et al. (2019). We evaluate a user simulator indirectly using the average success rate of the system agent trained by this simulator. It is called cross-model evaluation (Schatzmann & Young, 2009) which assumes a strategy learned with a good user model still performs well when tested on poor user models. It can indirectly evaluate the goodness of a user simulator. For direct evaluations, we adopt six evaluation measures to evaluate the diversity of user simulators automatically: average utterance length, vocabulary size, Dist-1, Dist-2 (Li et al., 2016a) and Entropy (Zhang et al., 2018). We also ask human users to rate the simulated dialogues [8] to assess the user simulators directly. We use five same metrics as Shi et al. (2019) which are Fluency, Coherence, Goal Adherence, Diversity, and Overall quality to assess user simulators from multiple aspects.

## G.3 TRAINING DETAILS OF USER SIMULATORS

We implement our GPT-based user simulators with DistilGPT2 (Sanh et al., 2020), a distilled version of GPT-2 by HuggingFace's Transformers (Wolf et al., 2020). We select the best performing models on the validation set through hyperparameters search of learning rate and batch size. The best models were fine-tuned with a batch size of 64 and a learning rate of 1e-3 over the corresponding dataset. We use the greedy decoding strategy for generating word-tokens in the inference phrase.

## G.4 EXPERIMENTS

**GPT-RNN.** Because the implementation of user simulator RNN mainly consists of NLU and NLG, we remove the POL module from U-GPT and use the same annotated data as RNN to fine-tune it to compare our U-GPT with the RNN-based methods fairly and name it as GPT-RNN.

As Tab. 11, Tab. 12, Tab. 13 show, GPT-RNN outperforms the user simulator RNN. It proves the power of leveraging GPT.

---

[8]The system agent for simulating dialogues is a third-party system provided by Shi et al. (2019) which was built based on hand-crafted rules.

Our GPT-RNN performs better than the user simulator RNNT, which can be seen from the cross-model evaluation results in Tab. 11, the automatic evaluation results in Tab. 12, and the Hu.Div score in the human evaluation results in Tab. 13. However, as Tab. 13 shows, RNNT performs better than our GPT-RNN in the overall performance from the human evaluation. We think this might be because (1) the third-party system also has an impact on the generated dialogues and (2) the NLG module of RNNT is the template-based method which leads to the generated dialogues from RNNT being easy for the third-party system to understand and interact with.

The automatic evaluation results in Tab. 12 and the Hu.Div score in the human evaluation results in Tab. 13 show that RNNR can generate more diverse language than our GPT-RNN. We think it is because the user utterances generated by RNNR are retrieved from a corpus that is written by real humans and the sentences written by humans are usually more diverse than the sentences generated by generative models. Even though the dialogues generated by RNNR are more diverse, the dialogues generated by our GPT-RNN are more fluent and coherent. Also, the cross-model evaluation results in Tab. 11 show that GPT-RNN can help to learn a more robust system agent than RNNR, but the Hu.All score in the human evaluation in Tab. 13 gives the opposite result.

Table 11: Cross study results. Each entry shows the success rate obtained by having the user simulator interacting with the RL system for 200 times.

| System \ User | AgenT | AgenR | AgenG | RNNT | RNNR | RNN | GPT | GPT$_{IL}$ | Avg.↑ | Std.↓ |
|---|---|---|---|---|---|---|---|---|---|---|
| Sys-RNNT | 30.5 | 23.0 | 35.5 | 99.0 | 97.5 | 84.0 | 75.5 | 66.0 | 63.9 | 28.5 |
| Sys-RNNR | 30.0 | 23.0 | 30.0 | 96.5 | 93.5 | 70.5 | 68.5 | 56.0 | 58.5 | 26.7 |
| Sys-RNN | 20.0 | 23.5 | 20.0 | 73.0 | 63.0 | 77.0 | 56.5 | 45.0 | 47.3 | **22.2** |
| Sys-GPT-RNN | 36.5 | 38.0 | 42.0 | 95.5 | 94.0 | 89.0 | 80.5 | 61.0 | **67.1** | 24.1 |

Table 12: Automatic evaluation results of RNNT, RNNR and GPT-RNN. The metrics include average utterance length (Utt), vocabulary size (Vocab), distinct-n (DIST-n) and entropy (ENT-n).

| User Simulators | Utt ↑ | Vocab ↑ | DIST-1 ↑ | DIST-2 ↑ | ENT-4 ↑ |
|---|---|---|---|---|---|
| RNNT | 9.83 | 192 | 0.77% | 1.51% | 4.24 |
| RNNR | 11.06 | 346 | 2.45% | 9.59% | 6.59 |
| RNN | 10.95 | 205 | 1.17% | 3.14% | 4.98 |
| GPT-RNN | 14.00 | 262 | 1.13% | 3.53% | 5.62 |

Table 13: Human evaluation results of RNNT, RNNR and GPT-RNN. The metrics include sentence fluency (Hu.Fl), coherence (Hu.Co), goal adherence (Hu.Go), language diversity (Hu.Div) and an overall score (Hu.All).

| User Simulators | Hu.Fl ↑ | Hu.Co ↑ | Hu.Go ↑ | Hu.Div ↑ | Hu.All ↑ |
|---|---|---|---|---|---|
| RNNT | 4.60 | 4.68 | 4.96 | 3.34 | 4.70 |
| RNNR | 3.92 | 3.88 | 4.72 | 3.94 | 4.16 |
| RNN | 2.80 | 2.30 | 2.86 | 2.74 | 2.30 |
| GPT-RNN | 4.10 | 4.04 | 4.30 | 3.70 | 4.00 |

# H    RELATIONS TO OTHER WORKS

**The inspiration behind our proposed MUST$_{adaptive}$ is similar to Focal Loss (Lin et al., 2017) and some curriculum learning approaches for reinforcement learning (Narvekar et al., 2020). Focal Loss addresses the extreme class imbalance by down-weighting the loss assigned to well-classified examples. Similarly, we assign lower weights to user simulators with which the dialogue system already performs well in the MUST$_{adaptive}$ training. However, we use a different way to obtain these weights. Curriculum learning approaches will define difficulty levels for different tasks. However, in our scenario, we cannot know how difficult each user simulator can be adapted beforehand. Moreover, curriculum learning approaches for reinforcement learning also move an agent from one task to another like the MUST$_{CRL}$ strategy.**

