# OpenReview forum: "One cannot stand for everyone! Leveraging Multiple User Simulators to train Task-oriented Dialogue Systems"
_ICLR.cc/2023/Conference — Submitted to ICLR 2023_

### Official Review · Reviewer_44kw · 2022-10-18

**Confidence:** 4
**Correctness:** 4
**Technical Novelty And Significance:** 3
**Empirical Novelty And Significance:** 3
**Recommendation:** 8

**Clarity, Quality, Novelty And Reproducibility:**

The idea is novel, but clarity and quality could be improved (see comments above for more details)

**Strength And Weaknesses:**

# Sumary of strengths

* To the best of my knowledge, the idea of combining multiple user simulators by casting this problem as a multi-armed bandit is novel.
* The way the idea is executed is not optimal (experiments on limited dataset), but the authors acknowledge this shortcoming. Despite the limitations of the paper's empirical grounding, its ideas and the evidence it does could provide open interesting avenues for future research on the interaction between user and agent simulators.
* The paper provides directional evidence showing that combining several user simulators is preferable to not doing so. This is reflected both in terms of overall success rate as measured automatically and by humans, and by showing that models trained through this method are less sensitive to unseen user simulators.
* It also evidences that adapting the distribution guiding the choice of user simulator during training results in faster training convergence.

# Summary of weaknesses

* There are some paragraphs that are poorly written, and make some key concepts difficult to understand.
* There are some conceptual gaps that make me doubt about the overall rigor of the paper.
* The more complex $MUST_{adaptive}$ model is only slightly better than the simpler $MUST_{uniform}$ baseline as measured by the in-domain automatic and human evaluations.


# Suggestions and questions for the authors.

* Good work! Do you intend to release the code in case the paper is accepted?
* In the second paragraph of page 2 you mention that "Extensive experimental results [...] show that the dialogue system trained by our proposed MUST achieves a better performance than those trained by any single user simulator", however I feel like the amount of experimentation described is not extensive, so I would suggest toning down this statement.
* In page 4 you introduce $MUST_{CRL}$ but you don't include that model in any of your results. In the paragraph "Challenges to leverage multiple user simulators" you mention some challenges related to this model. Are these challenges the reason why you did not analyze it? If that is the case then I suggest not giving it its own item (II) above, and instead give an overview of the "adaptive" variant of your framework. If this is not the case, then why was it not included?
* In the same "Challenges to leverage multiple user simulators." paragraph you mention: "unnecessary efforts will be costed for easily-adapted user simulators". I suggest rephrasing this statement as I don't think it's understandable in its current form.
* In the first paragraph of page 5 you link the "uniform adaptation" to reducing the catastrophic forgetting issue. However there is no experiment providing evidence that without "uniform forgetting" there's catastrophic forgetting. I suggest rephrasing this, or providing evidence that this is actually the case.
* In the first paragraph of section 5.2.2 you mention: "$SyS-MUST_{merging}$ is trained by $GPT_{IL}$ for implementing $MUST_{merging}$ strategy". First of all the statement is redundant by mentioning the "merging" strategy twice, but more importantly, I had understood that the merging strategy implied sampling dialogues from a set of user simulators and use these to train the final model. Does this statement mean that you generated dialogues with $GPT_{IL}$, which was trained on dialogues produced by 3 agenda-based simulators and MultiWoZ restaurant data, and define this method as sampling dialogues from several simulators?
* In the "Automatic Evaluation" paragraph of section 5.3 you say that the reason for the "merging" strategy not performing as well as the "uniform" and "adaptive" could be "because the merging strategy cannot effectively leverage multiple user simulators" which is a cyclical explanation. I suggest removing it or providing a more plausible explanation.
* In the third paragraph of section 5.3 you mention that the "uniform" and "adaptive" strategies achieve 2.4 absolute value improvements", but improvement over what?
* Finally, I suggest providing more details about the human evaluation. In Appendix B.2 you give some more details and mention that you "tell them how to judge if the generated dialogue is successful", "them" being the evaluators. But what was your definition of a successful dialogue? Every slot filled, for example? or something else?



# Typos and minor corrections

* Page 1, paragraph 4, line 3: best-performed -> best-performing
* P. 2, p. 1, l. 10: You reference challenges $\textit{i}$ and $\textit{ii}$ but you used 1 and 2 in the same paragraph for describing challenges. In the abstract you use $\textit{i}$ and $\textit{ii}$ for referring to the two types of adaptation rather than the challenges they address. I suggest being consistent with numbering.
* P. 2, p. 2, l. 5: Here you mention "$MUST_{adaptive}$ is indeed more efficient for leveraging multiple user simulators by our visualization analysis", which makes me wonder: more efficient than what? and what visualization analysis do you mean? I suggest clarifying this in the paper.
* P. 2, p. 3, l. 8: convergences -> converges
* P. 2, p. 6, l. 4: if accomplishing -> if it is accomplishing
* P. 3, p. 1, l. 1: Once the database result -> When the database result
* P. 4, p. 2, l. 1: first sample -> first samples
* P. 4, p. 2, l. 4: by RL algorithms -> with RL
* P. 4, p. 4, l. 11: Not sure what is meant by "unnecessary efforts will be costed for easily-adapted user simulators". I suggest rephrasing this.
* P. 4, p. 6, l. 1: recalls us a similar thought -> reminds us of a similar concept
* P. 4, p. 6, l. 3: weakly-performed -> weakly-performing; well-performed -> well-performing
* P. 4, p. 6, l. 4: "should reduce the interaction with user simulators that dialogue system has performed well and allocate more interactions with those user simulators that dialogue system has not performed well" -> "should reduce the interaction with user simulators with which the dialogue system has performed well and increase interactions in the opposite case."
* P. 5, p. 2, l. 5: masker's -> maker's
* P. 5, p. 4, l. 7: "p is expected to assign lower weights to user simulators that the system agent S already performs well and higher weights to those user simulators that S performs not well" -> "p is expected to assign lower weights to user simulators with which the system agent S already performs well and higher weights to those user simulators with which S performs poorly".
* P. 5, p. 7, l. 2: what do you mean with "latter" and "former" terms? Do you mean $\underbrace{\bar{x}_j}_\{\text\{exploitation\}\} + \underbrace{\sqrt{\frac{2\ln t}{T_\{j,t\}}}}_\{\text\{exploration\}\}$ in equation 2?
* P. 5, p. 8, l. 2: has been interacted so far -> has been interacted with so far
* P. 5, footnote, l. 2: "Then the index of the arm will be played from t = K + 1 to T is the sum of two terms: ..." makes no grammatical sense. Please correct it.
* P. 6, Algorithm 1: I suggest changing the verbs from -ing form to imperative, so initializing -> initialize, synthesizing -> synthesize, using -> use, evaluating -> evaluate, updating -> update. I also suggest clarifying what the lowercase s mentioned in the input is used for.
* P. 6, p. 1, l. 2-3: I think here you used both $\tau$ and $s$ to refer to the smoothing factor for distribution $\boldsymbol{p}$
* P. 6, p. 3, l. 3: "that the dialogue system has performed well" -> "with which the dialogue system has performed well"
* P. 7, p. 4, l. 4: "test the systems by them" -> "test the systems with them"
* P. 7, p. 4, l. 5: "there usually has a gap" -> "there usually is a gap"
* P. 9, Figure 2: There's no need to write "Tested by" for each subfigure. The name of the user simulator is enough. Also the label for (a) seems to be wrong.
* P. 14: There's mention to U-GPT here, but nowhere in the main text.


**Summary Of The Paper:**

This paper introduces the MUST framework (multiple user simulators), for training Task-oriented Dialogue systems with Reinforcement Learning and incorporating multiple user simulators.

The usual way to train these models is with a single system and user simulators but, as the authors correctly point out, a single user simulator might fail to exhaustively capture all possible human behaviors. This paper contributes a stepping stone towards this objective by casting the problem of combining different user simulators as a multi-armed bandit problem. The authors show that this method provides better results than using simpler combination baselines or single user simulators during training.


**Summary Of The Review:**

I think the idea is interesting, and that it could inspire interesting follow ups, however the execution leaves a bit to be desired. I will set my score to "marginally above the acceptance threshold" due to the lacking points I mentioned in detail above.

---

> ### Author Response · Authors · 2022-11-18
> **Replies to Reviewer 44kw : the second part**
>
> **Comment 3: In the first paragraph of page 5 you link the "uniform adaptation" to reducing the catastrophic forgetting issue. However there is no experiment providing evidence that without "uniform forgetting" there's catastrophic forgetting. I suggest rephrasing this, or providing evidence that this is actually the case.**
>
> Sorry for confusing you with "uniform adaptation". We refer it to as giving chances to all user simulators to interact with the system agent in a single RL environment as uniform adaptation.
> * $\mathrm{MUST}\_{\mathrm{CRL}}$ is a case that has no uniform adaptation for training the system agent. Our answer to Comment 2 shows that it indeed has the catastrophic forgetting issue.
> * $\mathrm{MUST}\_{\mathrm{adaptive}}$ and $\mathrm{MUST}\_{\mathrm{uniform}}$ are two cases that have uniform adaptation. Especially, $\mathrm{MUST}\_{\mathrm{uniform}}$ gives the same chances to all user simulators to interact with the system agent.
>
> The overall performance of Sys-$\mathrm{MUST}\_{\mathrm{adaptive}}$ and Sys-$\mathrm{MUST}\_{\mathrm{uniform}}$ (**the average success rates of 92.4 and 92.9 shown in Tab. 2**), compared with Sys-$\mathrm{MUST}\_{\mathrm{CRL}}$, provides evidence that the uniform adaptation can reduce catastrophic forgetting. We would clarify this point in the revision.
>
> -------------------------------------------------------------------------------------------------
>
> **Comment 4: Finally, I suggest providing more details about the human evaluation. In Appendix B.2 you give some more details and mention that you "tell them how to judge if the generated dialogue is successful", "them" being the evaluators. But what was your definition of a successful dialogue? Every slot filled, for example? or something else?**
>
> Sorry for the missing details about human evaluations. The criteria to judge whether a task-oriented dialogue is successful are based on two aspects:
> -  1) the system agent should correctly understand the user's goal (i.e., the predicted dialogue state tracking result is correct);
> -  2) the system agent should provide all information (i.e., all slot values or a booking reference number) that the user requests.
>
> **For human evaluations, we follow these standard criteria**. Besides, we also see if the system act generated by the system agent is matched to the user act for each turn in the dialogue. We have updated these details in App. B.5 of our revised paper.
>
> -------------------------------------------------------------------------------------------------
>
> **Comment 5: In the third paragraph of section 5.3 you mention that the "uniform" and "adaptive" strategies achieve 2.4 absolute value improvements", but improvement over what?**
>
> In **out-of-domain evaluation** where the user simulators used for testing are unseen by our MUST, Sys-$\mathrm{MUST}\_{\mathrm{uniform}}$ and Sys-$\mathrm{MUST}\_{\mathrm{adaptive}}$ **achieve at most 2.4 absolute value improvement over Sys-AgenR**. We want to emphasize that the dialogue systems trained with our MUST has a better generalization ability for interacting with unseen user simulators.
>
> -------------------------------------------------------------------------------------------------
>
> **Comment 6: Good work! Do you intend to release the code in case the paper is accepted?**
>
> Thanks for your encouragement.
> Yes, we have released the codes in the supplementary materials no matter whether the paper would be accepted or not.
>
> -------------------------------------------------------------------------------------------------
>
> **Comment 7: Writing suggestions.**
>
> We have fixed all typos as suggested. Thank you very much!

---

> ### Author Response · Authors · 2022-11-18
> **Replies to Reviewer 44kw : the first part**
>
> Thanks for your useful comments.
>
> **Comment 1: The more complex $\mathrm{MUST}\_{\mathrm{adaptive}}$ model is only slightly better than the simpler $\mathrm{MUST}\_{\mathrm{uniform}}$ baseline as measured by the in-domain automatic and human evaluations.**
>
> **The main difference** between $\mathrm{MUST}\_{\mathrm{adaptive}}$ and $\mathrm{MUST}\_{\mathrm{uniform}}$ is **training efficiency**. We provides **a visualization analysis in Fig. 2** for comparing them on convergence speeds and want to claim that **the proposed method $\mathrm{MUST}\_{\mathrm{adaptive}}$ can converge faster than $\mathrm{MUST}\_{\mathrm{uniform}}$**. This benefit mainly owes to that the $\mathrm{MUST}\_{\mathrm{adaptive}}$ strategy can adaptively adjust the weights of different user simulators according to the current performance of the trained system agent.
>
> -------------------------------------------------------------------------------------------------
>
> **Comment 2: In page 4 you introduce $\mathrm{MUST}_{\mathrm{CRL}}$ but you don't include that model in any of your results. In the paragraph "Challenges to leverage multiple user simulators" you mention some challenges related to this model. Are these challenges the reason why you did not analyze it? If that is the case then I suggest not giving it its own item (II) above, and instead give an overview of the "adaptive" variant of your framework. If this is not the case, then why was it not included?**
>
> Yes, we did not analyze the $\mathrm{MUST}\_{\mathrm{CRL}}$ strategy are mainly **because it has a problem of catastrophic forgetting and would be sensitive to the order of different user agents interacting with the dialogue system**. As suggested by the reviewer, we have added some experiments on $\mathrm{MUST}_{\mathrm{CRL}}$ (**See App. C. in our revised paper for more details**).
>
> Without losing any generality, we consider two representative sequential orders:
> 1) AgenT, AgenR, RNNT, GPT; and
> 2) AgenR, GPT, AgenT, RNNT.
>
> For case 1, the first two user simulators are Agenda-based user simulators; the last two user simulators are Neural networks-based user simulators. For case 2, we interleave these two types of user simulators. When the system trained by a user simulator converges, we let it continue to interact with another user simulator following the order.
>
> The experimental results of **Case 1** (i.e., AgenT, AgenR, RNNT, GPT) are shown as follows:
>
> |System agents \ User simulators  | tested with AgenT    | tested with AgenR  | tested with RNNT    | tested with GPT     | Avg.    |
> |---------------------------      |:--------:| :-----:|:-------:|:-------:| -------:|
> |the system trained by AgenT      | 97.5     | 54.0   |  98.5   |  78.0   | 82.0 |
> |the system trained by AgenT, AgenR sequentially            | 97.0  | 93.0   |  97.0  |  82.5 |**92.4**|
> |the system trained by AgenT, AgenR, RNNT sequentially      | 95.0  |**59.5**|  97.0  |  80.5 | 83.0|
> |the system trained by AgenT, AgenR, RNNT, GPT sequentially | **75.5** |**47.5**| 96.0 | 82.0 | 75.3|
>
> In case 1, the system agent achieves the best performance (i.e., 92.4 in terms of the average success rate) after training with AgenT and AgenR sequentially.
> * However, its overall performance **degrades to 83.0 after training with RNNT**; especially, its performance **decreases by 36.0\% when testing with AgenR ($93.0 \rightarrow 59.5$)**.
> * Moreover, after continuing to learn from GPT, the performance of the system agent becomes worse for AgenT ($95.0 \rightarrow 75.5$) and AgenR ($59.5 \rightarrow 47.5$).
>
> The experimental results of **Case 2** (i.e., AgenR, GPT, AgenT, RNNT) are shown as follows:
>
> |System agents \ User simulators | tested with AgenT  |tested with AgenR  | tested with RNNT   | tested with GPT| Avg.|
> |---------------------------   |:------:| :-----:|:-------:| :-------:| -------:|
> |the system trained by AgenR                   | 96.0   | 90.0   |  98.5   |  82.5   | **91.8**|
> |the system trained by AgenR, GPT sequentially            | 97.5   | 88.0   |  97.0   |  81.5   | 91.0|
> |the system trained by AgenR, GPT, AgenT sequentially      | 96.5  |**78.5**|  97.0   |  80.0   | 88.0|
> |the system trained by AgenR, GPT, AgenT, RNNT sequentially | 97.5 |**65.5**|  95.0   |  78.5   | 84.1|
> * In case 2, we can also see that the catastrophic forgetting issue heavily happened when the system agent starts learning from AgenR ($88.0 \rightarrow 78.5 \rightarrow 65.5$).
>
> **These results can confirm that implementing our proposed MUST with the $\mathrm{MUST}\_{\mathrm{CRL}}$ strategy indeed has the catastrophic forgetting issue.**
>
> -------------------------------------------------------------------------------------------------

---

### Official Review · Reviewer_mwD2 · 2022-10-23

**Confidence:** 3
**Correctness:** 3
**Technical Novelty And Significance:** 2
**Empirical Novelty And Significance:** 3
**Recommendation:** 6

**Clarity, Quality, Novelty And Reproducibility:**

Clarity: No problems with clarity.

Quality: The experiments are set up well but the theses would be better served by more comprehensive experiments.

Novelty: The approach is founded in bandit literature but advances beyond prior usages of multiple user simulators in this field.

Reproducibility: Code and data are provided in supplement, but I have not verified their accuracy or usability.

**Strength And Weaknesses:**

Strengths:
- MUST as an adaptive approach seems well-justified
- Discussion of learning curves motivates further study of MUST as a possible adaptive approach for low data regimes
- Experiments explicitly target enumerated challenges with existing user simulators and mixtures (catastrophic forgetting, over-fitting to specific simulators)

Weaknesses:
- It is important to see empirically if MUST generalizes to a larger or smaller pool of user simulators and is not over-tuned for the current set chosen in the paper. For example, ablation studies with different subsets of the user simulators would be helpful
- Needs more discussion of the limitations on user behavior diversity etc. given that only the Restaurant domain is evaluated here. It is unclear from the experiments here how well the approach can succeed across multiple domains where the pool of user simulators potentially explodes
- Human evaluation details are relatively scarce - would like to see justification for having human raters apparently rate the success rate (the same as the automatic evaluation metric), rather than other metrics of quality (even subjective quality rating) that leverages human annotators more effectively

**Summary Of The Paper:**

The authors introduce a multi-armed bandit approach to training task-oriented dialog (TOD) systems to leverage multiple user simulators. This approach is demonstrated to outperform ablative variants that suffer from catastrophic forgetting and over-fitting to specific "easy" simulators.

**Summary Of The Review:**

The authors propose a well-justified bandit based method for learning from mixtures of different user simulators to train a TOD model. Automatic evaluation is encouraging but the central assertion of adaptibility probably requires a more comprehensive evaluation setting (models, domains)

---

> ### Author Response · Authors · 2022-11-18
> **Replies to Reviewer mwD2 : the second part**
>
> **Comment 2: Needs more discussion of the limitations on user behavior diversity etc. given that only the Restaurant domain is evaluated here. It is unclear from the experiments here how well the approach can succeed across multiple domains where the pool of user simulators potentially explodes**
>
>
> **Limitation on domain diversity**
> We admit that only validating our MUST on the Restaurant domain of the MultiWOZ dataset is the main limitation of our work, see the conclusion section. There is no previous work that considers leveraging multiple user simulators simultaneously to train the system agent before. To validate our proposed MUST, we can only find some user simulators from Shi et al. (2019), while they build these simulators only on the restaurant search task.
>
> **Leveraging more user simulators**
> We have made some efforts to use more user simulators to validate our MUST, see more details in the newly-added App. F. The experiments show that Sys-$\mathrm{MUST}\_{\mathrm{adaptive}}$ converges faster than Sys-$\mathrm{MUST}\_{\mathrm{uniform}}$ when using eight user simulators, which is consistent with the settings when using three, four or five user simulators. The experiments evidence that the MUST training can still succeed when we increase the number of user simulators to eight.
>
>
> **The curriculum learning of multiple user simulators in MUST**
> We also compare the learning curves of Sys-$\mathrm{MUST}\_{\mathrm{adaptive}}$ and Sys-$\mathrm{MUST}\_{\mathrm{uniform}}$ trained with different numbers of user simulators (i.e., four, five, and eight user simulators). It is a fair comparison because all these groups of user simulators include the hardest user simulator AgenR and the easiest user simulator RNNT that can be adapted by the system (See Sec. 5.4). We observe that with more user simulators, Sys-$\mathrm{MUST}\_{\mathrm{adaptive}}$ not only performs better but also converges faster than with fewer user simulators. Moreover, it might be hard to converge if we train the system agent only with AgenR. We could use curriculum learning to explain this phenomenon.
>
> Therefore, we preliminarily find that
> 1) our MUST could be generalized to a large set of user simulators and
> 2) leveraging more user simulators (even from other domains) would be useful for the system's convergence.
>
> For multiple domain scenarios, it will be more difficult to manually build larger-scale and more diverse user simulators. Therefore, we will leave building a larger set of user simulators across multiple domains as future work since it is not the main focus of this paper.
>
> -------------------------------------------------------------------------------------------------
>
> **Comment 3: Human evaluation details are relatively scarce - would like to see justification for having human raters apparently rate the success rate (the same as the automatic evaluation metric), rather than other metrics of quality (even subjective quality rating) that leverages human annotators more effectively.**
>
> Sorry for the missing details about human evaluations.
> **Same as the automatic evaluation metric, our human evaluations mainly focus on the success rates of dialogue systems rather than other metrics of language quality.**
> The criteria to judge whether a task-oriented dialogue is successful are based on two aspects:
> -  1) the system agent should correctly understand the user's goal (i.e., the predicted dialogue state tracking results are correct);
> -  2) the system agent should provide all information (i.e., all slot values or a booking reference number) that the user requests.
>
> **For human evaluations, we follow these standard criteria.** Besides, we also see if the system act generated by the system agent is matched to the user act for each turn in the dialogue. We have updated these details in App. B.5 of our revised paper.

---

> ### Author Response · Authors · 2022-11-18
> **Replies to Reviewer mwD2 : the first part**
>
> **Comment 1: It is important to see empirically if MUST generalizes to a larger or smaller pool of user simulators and is not over-tuned for the current set chosen in the paper. For example, ablation studies with different subsets of the user simulators would be helpful**
>
> Thanks for your suggestions. We now consider another two groups of user simulators (OOD represents Out-of-domain evaluation) for ablation studies:
>   * Case 1: **Three user simulators**: AgenT, RNNT, GPT
>     |System agents \ User simulators  | AgenT  | RNNT  | GPT   | AgenR | AgenG | RNNR | RNN  | OOD Avg.|Avg. |
>     |---------------------------      |:------:| :----:|:-----:| -----:| :----:|:----:| ----:| ----:|----:|
>     |Sys-AgenT                        | 97.5     |  98.5   |  78.0   |  54.0 | 72.5 | 92.5 | 77.0 | 74.0|  81.4 |
>     |Sys-RNNT                         | 30.5     |  99.0   |  75.5   |  23.0 | 35.5 | 97.5 | 84.0 | 60.0|  63.6 |
>     |Sys-GPT                          | 60.5     |  97.0   |  82.0   |  51.5 | 59.5 | 94.0 | 92.0 | 74.3 | 76.6 |
>     |Sys-$\mathrm{MUST}_{\mathrm{uniform}}$  | 97.5  | 97.5 |  82.5  | 55.5 | 80.5 | 97.0 | 87.0 | 80.3 | **85.4** |
>     |Sys-$\mathrm{MUST}_{\mathrm{adaptive}}$   | 97.5  | 96.0 |  82.5  | 55.0 | 82.0 | 97.5 | 87.0 | 80.0 | **85.4** |
>
>   * Case 2: **Five user simulators**: AgenT, AgenR, RNNT, RNNR, GPT
>     |System agents \ User simulators  | AgenT  | AgenR | RNNT  | RNNR |  GPT  | AgenG| RNN  | OOD Avg. | Avg. |
>     |---------------------------      |:------:| :----:|:-----:| ----:| :----:|:----:| ----:| ----:|----:|
>     |Sys-AgenT                        | 97.5   |  54.0 |  98.5 | 92.5 |  78.0 | 72.5 | 77.0 | 74.8 | 81.4 |
>     |Sys-AgenR                        | 96.0   |  90.0 |  98.5 | 97.5 |  80.5 | 97.5 | 82.0 | 89.8 | 91.7 |
>     |Sys-RNNT                         | 30.5   |  23.0 |  99.0 | 97.5 |  75.5 | 35.5 | 84.0 | 59.8 | 63.6 |
>     |Sys-RNNR                         | 30.0   |  23.0 |  96.5 | 93.5 |  68.5 | 30.0 | 70.5 | 50.3 | 58.9 |
>     |Sys-GPT                          | 60.5   |  51.5 |  97.0 | 94.0 |  82.0 | 59.5 | 92.0 | 75.8 | 76.6 |
>     |Sys-$\mathrm{MUST}_{\mathrm{uniform}}$  | 97.0  | 89.0 |  97.0  | 97.5 | 82.5 | 97.5 | 87.5 | 91.8 | **92.6** |
>     |Sys-$\mathrm{MUST}_{\mathrm{adaptive}}$   | 97.5  | 87.0 |  97.0  | 97.5 | 82.0 | 96.5 | 87.0 | **92.5** |92.1 |
>
>
> From the two tables above, we can observe that Sys-$\mathrm{MUST}\_{\mathrm{uniform}}$ and Sys-$\mathrm{MUST}\_{\mathrm{adaptive}}$ largely outperform the dialogue systems trained by single user simulators in the overall performance. Especially, they **gain an improvement of 4 absolute points (85.4 vs. 81.4)** when trained with three user simulators of AgenT, RNNT, and GPT. In summary, MUST could **consistently improve** the performance of the systems when using different numbers of user simulators.
>
> **When testing our MUST with unseen user simulators**, Sys-$\mathrm{MUST}\_{\mathrm{uniform}}$ and Sys-$\mathrm{MUST}\_{\mathrm{adaptive}}$ can also largely outperform the dialogue systems trained by a single user simulator. Sys-$\mathrm{MUST}\_{\mathrm{adaptive}}$ achieves **a 2.7 absolute value improvement (92.5 vs 89.8) over Sys-AgenR** in case 2. Sys-$\mathrm{MUST}\_{\mathrm{uniform}}$ and Sys-$\mathrm{MUST}\_{\mathrm{adaptive}}$ even improve **at least 5.7 points (80.0 vs 74.3) over Sys-GPT** in case 1. These experimental results on different subsets of user simulators demonstrate that our MUST has **a better generalization ability** for interacting with unseen user simulators and is insensitive to the user simulator selection.
>
> When using three, four, or five user simulators to implement MUST, we could conclude that training the dialogue system by $\mathrm{MUST}\_{\mathrm{adaptive}}$ **consistently converges faster than** by $\mathrm{MUST}\_{\mathrm{uniform}}$ (see Fig. 4(a), Fig. 2(a), and Fig. 4(b) respectively).
>
> The ablation studies on different subsets of user simulators can demonstrate the superiority and robustness of our proposed MUST. We have put these ablation studies in App. D. of our revised paper.

---

### Official Review · Reviewer_H1xa · 2022-11-05

**Confidence:** 4
**Correctness:** 3
**Technical Novelty And Significance:** 2
**Empirical Novelty And Significance:** 2
**Recommendation:** 5

**Clarity, Quality, Novelty And Reproducibility:**

Overall, I think the novelty is limited, given the marginal performance improvement over baseline and the limited number of user simulators used for training.

**Strength And Weaknesses:**

Strength:

1. The idea of adapting multiple user simulators to train a dialogue policy is interesting. The adaptive phase is closely related to curriculum learning.
2. Instead of using the mean success rate to calculate sampling ratio, the authors proposed to use the UCB of success rate, which hopefully makes the algorithm more robust. Although there is no ablation study to show if this is true.

Weaknesses:

1. The authors only used 4 user simulators to train MUST_uniform and MUST_adaptive, which makes the work less impressive. In general, we wouldn't expect many rule-based user simulators exist, therefore, we would rely on data-driven user simulators. I think the work is more valuable if the authors extend the current work and learn a large set of (and possibly latent/complementary) user simulators from data, and then apply the proposed methods to see if the policy can benefit from the large pool of user simulators.

2. Follow up on the previous point, the authors didn't answer the question of how many user simulators are enough, and how different should they behave?

3. The related works should include curriculum learning literatures. A lot of works use the loss/rewards to adjust task/example weights. For example, on top of my head, focal loss[1] is one method. There should also be an ablation study to show that the UCB term is useful.

4. The authors proposed to use the inverse of UCB as the weights of user simulators. I am a little bit confused. The z_j in Section 4.2 Equation (3) is the *inverse* of success rate, reflecting the task difficulty. Therefore, we should use the UCB of z_j to calculate sampling ratios. In that case, should we pick lower confidence bound in equation (2)? For example, let's say we have two user simulators, the first one has a CI [0.2, 0.8], the second one has a CI [0.4, 0.6]. The proposed algorithm in the paper will assign lower weight to the first user simulator. However, the first user simulator has a lower bound of 0.2, which means that the user simulator could be very hard to train, and should be assigned a higher weight.

5. Based on the results in Table 2 and Table 3, The performance of MUST_uniform and MUST_adaptive are super close. The performance of MUST_adaptive and SYS_AgenR are also very close. I don't see a clear benefit of the proposed algorithm.

6. The experiments only use MultiWOZ restaurant domain dataset, and no SoTA methods are compared.

[1] Focal Loss for Dense Object Detection https://arxiv.org/abs/1708.02002

**Summary Of The Paper:**

The authors proposed to use multiple user simulators to train a dialogue policy and showed that using multiple user simulators rather than a single user simulator improves the policy performance.

The authors developed a sampling strategy to sample user simulators during training. The sampling ratios are inversely proportional to the policy's performance on each user simulator. The policy's performance is calculated by the success rate's upper confidence bound.

**Summary Of The Review:**

The authors proposed to use multiple user simulators to train a dialogue policy. While the idea is interesting, it is only trained on a small set of user simulators. The authors didn't justify the usefulness of the proposed UCB-based performance expectation. Finally, the performance improvements are marginal and are not compared with SoTA methods.

---

> ### Author Response · Authors · 2022-11-18
> **Replies to Reviewer H1xa (4/4):**
>
> **Comment 6: Based on the results in Table 2 and Table 3, The performance of MUST_uniform and MUST_adaptive are super close. The performance of MUST_adaptive and SYS_AgenR are also very close. I don't see a clear benefit of the proposed algorithm.**
>
>
> * **The main difference** between $\mathrm{MUST}\_{\mathrm{adaptive}}$ and $\mathrm{MUST}\_{\mathrm{uniform}}$ is **training efficiency**. We provide **a visualization analysis in Fig. 2** for comparing them on convergence speeds and want to claim that **the proposed method $\mathrm{MUST}\_{\mathrm{adaptive}}$ can converge faster than $\mathrm{MUST}\_{\mathrm{uniform}}$**. This benefit mainly owes to the $\mathrm{MUST}\_{\mathrm{adaptive}}$ strategy can adaptively adjust the weights of different user simulators according to the current performance of the trained system agent.
>
> * Even if the system agent trained by AgenR is the previous SOTA system, Sys-$\mathrm{MUST}\_{\mathrm{adaptive}}$ has a 1.2 absolute value improvement (92.9 vs. 91.7) over Sys-AgenR averagely. Moreover, **when testing with the user simulators that are unseen by our MUST, Sys-$\mathrm{MUST}_{\mathrm{adaptive}}$ achieve 2.4 absolute value improvement over Sys-AgenR** (see Tab. 2 in our paper). This evidences that MUST has a better generalization ability for interacting with unseen user simulators.
>
> We have made it clear in the revised version.
>
>
> ----------------------------------------------------------------------------------
>
> **Comment 7: The experiments only use MultiWOZ restaurant domain dataset, and no SoTA methods are compared.**
>
> **Only use the MultiWOZ restaurant domain**
> There is no previous work that considers leveraging multiple user simulators simultaneously to train the dialogue system. To validate our proposed MUST, we can only find some user simulators from Shi et al. (2019), while they build these simulators only on the restaurant search task.  We admit that this is the main limitation of our work, see the conclusion section.
>
> The current SoTA methods on the MultiWOZ dataset did not report the performance of their methods on the **restaurant domain**.  It is not easy to test their methods on the restaurant domain because the system acts are re-annotated by Shi et al. (2019) which leads to an annotation mismatch between our systems and the current SoTA methods on the MultiWOZ dataset.
>
> Typically, researchers use the static test dataset to evaluate the performance of the methods that build the ToD systems. However, it might be more reasonable to let the system agent interact with a user simulator to test its performance, which is why we adopt such a way to evaluate all dialogue systems in this paper.
>
>
> ----------------------------------------------------------------------------------
>
> **Comment 8: Instead of using the mean success rate to calculate sampling ratio, the authors proposed to use the UCB of success rate, which hopefully makes the algorithm more robust. Although there is no ablation study to show if this is true.**
>
> We have added some ablation studies in App. E. to validate the rationale and robustness of our modified UCB1 algorithm. Thanks for your constructive comment.

---

> ### Author Response · Authors · 2022-11-18
> **Replies to Reviewer H1xa (3/4):**
>
> **Comment 5: The authors proposed to use the inverse of UCB as the weights of user simulators. I am a little bit confused. The z_j in Section 4.2 Equation (3) is the inverse of success rate, reflecting the task difficulty. Therefore, we should use the UCB of z_j to calculate sampling ratios.
> In that case, should we pick lower confidence bound in equation (2)? For example, let's say we have two user simulators, the first one has a CI [0.2, 0.8], the second one has a CI [0.4, 0.6]. The proposed algorithm in the paper will assign lower weight to the first user simulator. However, the first user simulator has a lower bound of 0.2, which means that the user simulator could be very hard to train, and should be assigned a higher weight.**
>
> **We reinterpret your concern as explaining the rationale** why we combine the exploitation term and the exploration term in their current form to get the distribution $p$. Please correct us if we misunderstood your concern.
>
> Since each $\bar x_j$ ($\bar x_j$ is the success rate of the system agent tested with user simulator $U_j$) is smaller than 1, then $\frac{1}{\bar x_j}$ will be larger than 1. We think that the term of $\frac{1}{\bar x_j}$ and the exploration term of $\sqrt{\frac{2\ln t}{T_{j,t}}}$ (also smaller than 1) are likely not of the same magnitude, which will lead to a consequence that the exploitation term becomes dominant while the exploration term is negligible. Therefore, we design the distribution $p$ as the current form (Eq. 2, Eq. 3, and Eq. 4 in Sec. 4.2).
>
> Moreover, **the exploration (UCB) term in our modified UCB1 algorithm does not measure the uncertainty of a user simulator anymore since the algorithm has a warmup phase**, which will select each user simulator many times not just select once. The exploration term mainly plays a role of "uniform adaptation", which can reduce the catastrophic forgetting issue.
>
> **There have several different ways to obtain the distribution $p$, see more details in App. E.2**. We conduct experiments to compare the performances of these different ways and the results are shown in Fig. 5(b). For the convergence speeds of training the system agent, case 1 is worse than our current version in the paper and case 2 and case 3 are comparative to our current version.
>
> |variants                   | exploitation-exploration term  | post-processing   |  distribution|
> |--------------------------------------|:------| :----|:-----:|
> |$\mathrm{MUST}\_{\mathrm{adaptive}}$  | $\hat x_j=\bar x_j + \sqrt{\frac{2\ln t}{T_{j,t}}}$  |  $z_j = \frac{1}{ \left({\hat x_j - \tau \min(\{ \bar x_1, \cdots , \bar x_K\})} \right)}$ | $p_j = \frac{z_j}{\sum_{i=1}^{K} z_i}$ |
> |w/o exploration                       | $ z_j = \frac{1}{\bar x_j} $  |  |  $p_j = \frac{z_j}{\sum_{i=1}^{K} z_i}$ |
> |$\mathrm{MUST}\_{\mathrm{adaptive-I}}$| $\hat x_j = \frac{1}{\bar x_j} + \sqrt{\frac{2\ln t}{T_{j,t}}}$   |  $z_j=\hat x_j-\tau \min(\{ 1/\bar x_1, \cdots, 1/\bar x_K \})$ |  $p_j = \frac{z_j}{\sum_{i=1}^{K} z_i}$ |
> |$\mathrm{MUST}\_{\mathrm{adaptive-II}}$| $\hat x_j= \frac{1/{\bar x_j} }{\sum_{i=1}^{K} {1}/{\bar x_i} } $, $\hat z_j = \hat x_j +\sqrt{\frac{2\ln t}{T_{j,t}}}$ | $z_j=\hat z_j-\tau \min(\{ \hat x_1, \cdots , \hat x_K \})$ |  $p_j = \frac{z_j}{\sum_{i=1}^{K} z_i}$ |
> |$\mathrm{MUST}\_{\mathrm{adaptive-III}}$| $\hat x_j=\frac{1}{(\bar x_j-\tau \min(\{ \bar x_1, \cdots , \bar x_K \}))}$,  $z_j=\frac{\hat x_j}{\sum_{i=1}^{K} \hat x_i} + \sqrt{\frac{2\ln t}{T_{j,t}}}$  |  | $p_j = \frac{z_j}{\sum_{i=1}^{K} z_i}$ |
>
>
> **Results for ablation study on the variants.** Experimental results of these different variants are shown in Fig. 5.  The convergence speed of $\mathrm{MUST}\_{\mathrm{adaptive}}$-I is much slower compared to others, which demonstrates that the exploration term is useful once more.
> The convergence speeds of $\mathrm{MUST}\_{\mathrm{adaptive}}$-II and $\mathrm{MUST}\_{\mathrm{adaptive}}$-III are comparative to $\mathrm{MUST}\_{\mathrm{adaptive}}$. This probably shows that our designed  $\mathrm{MUST}\_{\mathrm{adaptive}}$ with three operations (i.e., exploitation-exploration term construction, re-scaling strategy, and the inversion of $\bar x_j$) is not only reasonable but also robust to the order permutation of these three operations.

---

> ### Author Response · Authors · 2022-11-18
> **Replies to Reviewer H1xa (2/4):**
>
> **Comment 3: The related works should include curriculum learning literatures. A lot of works use the loss/rewards to adjust task/example weights. For example, on top of my head, focal loss[1] is one method.**
>
> Thanks for your suggestions. Curriculum learning and focal loss indeed are related to our work, and we have cited some literature about them in our paper, see the last paragraph of Sec. 4 and App. H.
>
> Focal Loss addresses the extreme class imbalance by down-weighting the loss assigned to well-classified examples. Similarly, we assign lower weights to user simulators with which the dialogue system already performs well in the $\mathrm{MUST}\_{\mathrm{adaptive}}$ training. However, we use a different way to obtain these weights. Curriculum learning approaches will define difficulty levels for different tasks. However, in our scenario, we cannot know how difficult each user simulator can be adapted beforehand. Moreover, curriculum learning approaches for reinforcement learning also move an agent from one task to another like the $\mathrm{MUST}\_{\mathrm{CRL}}$ strategy.
>
>
>
> -----------------------------------------------------------------------------------
>
> **Comment 4: About the ablation study to show that the UCB term is useful.**
>
> **We have added some experimental results for the ablation study on the UCB term, see these details in App. E.1 of our revised paper.**
> Our **modified** UCB1 algorithm provides a distribution $p$ for guiding how to sample different user simulators to accelerate the entire MUST training. The exploration term in the proposed $\mathrm{MUST}\_{\mathrm{adaptive}}$ exists mainly for uniform adaption (see the detailed explanation in Sec. 4.1). We conduct some ablation studies to evidence the necessity of the exploration term.
>
> **$\mathrm{MUST}\_{\mathrm{adaptive}}$ w/t exploration**
> If we omit the exploration term in our **modified** UCB1 algorithm, the simplest way to calculate the distribution $p$ is to make the sample probability w.r.t a user simulator solely depend on the inversion of the system's performance. See the row called `w/t exploration' in Tab. 10 for comparisons.
>
> In this situation, the obtained distribution $p$ might be sharp due to the lack of the exploration term, which would be harmful for uniform adaption to some extent. As Fig. 6(a) shows, $\mathrm{MUST}\_{\mathrm{adaptive}}$ performs worse and converges slower when omitting the exploration term, compared with when our **modified** UCB1 algorithm has the exploration term. This could demonstrate both the importance of uniform adaption and the usefulness of the exploration term.

---

> ### Author Response · Authors · 2022-11-18
> **Replies to Reviewer H1xa (1/4):**
>
> **Comment 1: The authors only used 4 user simulators to train MUST_uniform and MUST_adaptive, which makes the work less impressive. In general, we wouldn't expect many rule-based user simulators exist, therefore, we would rely on data-driven user simulators. I think the work is more valuable if the authors extend the current work and learn a large set of (and possibly latent/complementary) user simulators from data, and then apply the proposed methods to see if the policy can benefit from the large pool of user simulators.**
>
> Thanks for your suggestion on leveraging a large set of user simulators, which is also what we want to explore.
> Although the old version of this paper only involves only a few user simulators, these simulators are representative; some of them are rule-based and some of them are data-driven.
>
> We conduct experiments to use more user simulators to validate our MUST, see more details in the newly-added App. F. The experiments show that Sys-$\mathrm{MUST}\_{\mathrm{adaptive}}$ converges faster than Sys-$\mathrm{MUST}\_{\mathrm{uniform}}$ when using eight user simulators (six of them are data-driven user simulators), which is consistent with the settings when using three, four or five user simulators. The experiments evidence that the MUST training can still succeed when we increase the number of user simulators to eight.
>
> We also compare the learning curves of Sys-$\mathrm{MUST}\_{\mathrm{adaptive}}$ and Sys-$\mathrm{MUST}\_{\mathrm{uniform}}$ trained with different numbers of user simulators (i.e., four, five, and eight user simulators). It is a fair comparison because all these groups of user simulators include the hardest user simulator AgenR and the easiest user simulator RNNT that can be adapted by the system (See Sec. 5.4). We observe that, with more user simulators, Sys-\madp\ not only performs better but also converges faster than with fewer user simulators. Moreover, it might be hard to converge if we train the system agent only with AgenR. We could use curriculum learning to explain this phenomenon.
>
> Therefore, we preliminarily find that
> 1) our MUST could be generalized to a large set of user simulators and
> 2) leveraging more user simulators might be useful for the system's convergence.
>
> -----------------------------------------------------------------------------------
>
> **Comment 2: Follow up on the previous point, the authors didn't answer the question of how many user simulators are enough, and how different should they behave?**
>
> **how many user simulators are enough**
> Since the $\mathrm{MUST}_{\mathrm{adaptive}}$ is adaptive, at least in principle, we believe that it could deal with as many user simulators as possible. The more diversely these user simulators behave, the more beneficial it will be for MUST. However, in practice, some user simulators might be well-designed and the system might perform poorly with them, which will bring much time cost for the RL training. However, we believe that adding more user simulators to MUST would not be harmful to the system's performance, but it might make the RL training slightly slower.
>
> Since it is limited in simulator design under the current scenario, we are unable to include a much larger set of user simulators, which we have to leave as future work.
>
>
> **how different should they behave**
> We believe that *the more diversely they behave, the more beneficial it will be for MUST*. However, it is not easy to measure how differently they behave quantitatively since some of them are rule-based and some are data-driven.
> Ideally, if the dialogue state space explored by the system agent is n-dimensional, there will need at least n user simulators that are orthogonal with each other, and their dialog acts can span the entire dialogue state space.

---

### Author Response · Authors · 2022-11-18
**General response to all reviewers:**

We thank the reviewers for recognizing our idea is novel (**44kw**), well-justified (**mwD2**), and interesting (**H1xa**).  Some reviewers also agreed that this work could open interesting avenues for future research on the interactions between user agents and system agents (**44kw**) and might be inspirable for adaptive approaches for low data regimes (**mwD2**).

We would like to highlight some revisions that might improve the general quality of this paper.
- We conducted some additional experiments to check the performance of the dialogue systems trained by MUST with different subsets of user simulators (i.e., three and five user simulators). The results demonstrated the superiority and robustness of our proposed MUST, see App. D.
- We conducted ablation studies for the modified UCB1 algorithm. We first showed that the exploration (UCB) term is useful from the experimental results. Then we illustrated several different variants to modify the UCB1 algorithm for obtaining the distribution $p$ under the $\mathrm{MUST}_{\mathrm{adaptive}}$. Finally, our experiments validated the reasonability of our modified UCB1 algorithm, see App. E.
- We have made some efforts to build more user simulators to validate our MUST,  see App. F. The experiments showed that $\mathrm{MUST}\_{\mathrm{adaptive}}$ converges faster than $\mathrm{MUST}\_{\mathrm{uniform}}$ when using eight user simulators, which is consistent with the settings when using three, four or five user simulators. The experiments preliminarily showed that our MUST could be generalized to a larger set of user simulators. We leave leveraging a larger pool of user simulators as future work.
- We implemented our MUST with the $\mathrm{MUST}\_{\mathrm{CRL}}$ strategy and gave an analysis for the results. It confirmed that $\mathrm{MUST}_{\mathrm{CRL}}$ strategy has the problem of catastrophic forgetting, see App. C.

Best,

The authors

---

### Author Response · Authors · 2022-11-25
**If any clarification needed?**

Dear Reviewers,

Could you please let us know if we could further clarify or provide anything？

Thanks in advance.


The authors

---

### Decision · Program_Chairs · 2023-01-20

**Decision:**

Reject

**Justification For Why Not Higher Score:**

The core concern with this work is that they basically combine a small number of user simulators to work on a relatively small dataset -- demonstrating interesting, but thus preliminary findings. Without additional empirical and/or theoretical work, it isn't clear that this is a viable method that be broadly applied in building dialogue systems.

**Justification For Why Not Lower Score:**

N/A

**Metareview: Summary, Strengths And Weaknesses:**

The authors consider the problem of training a dialogue policy with a set of multiple user simulators, developing a multi-armed bandit motivated strategy for selecting between the user simulators at each turn. Specifically, they develop a sampling policy where the sampling distribution is such that the sampling rate for a user simulator is inversely proportional to the UCB of the estimated success rate. The resulting approach mitigates catastrophic forgetting and avoid overfitting through the proposed sampling strategy. Using both agenda-based and learned user simulators recently developed for other work [Shi, et al., EMNLP19] and the restaurant subset of MultiWoZ, they show improved performance over any single user simulator and that their adaptive sampling rate method empirically has the best convergence properties.

The consensus strengths of this work include:
- Adapting multiple user simulators to learn a policy is interesting and MAB-based approaches are sensible and conceptually appealing.
- The authors consider multiple sampling strategies that are used as baselines for the proposed adaptive setting (that is closely related to curriculum learning)
- Using the UCB is well-motivated/established within the MAB formulation for increasing robustness of the resulting MAB system.
- The experiments include a good 'dive deep' element, especially when considering the appendices (which address some of the issues in the initial reviews)

The weaknesses discussed in the reviews (and my own reading) of this work include:
- The work is more of a proof-of-concept since they use a small number of existing simulators and don't characterize the relationship between the user simulators. Furthermore, this is difficult to address in future work as creating multiple good user simulators is difficult for a given setting (maybe personas?) -- which calls into question if this is a viable method in the general case. Even after partially addressing in the rebuttal, it really isn't clear how to scale this based on the submitted work, especially in academic research settings on benchmark datasets.
- This work is specific to dialogue systems and conceptually is a relatively straightforward application of MAB; while novel in this specific setting, it is somewhat straightforward if one assumes the existence of multiple simulators. In the other direction, there aren't any findings that contribute to general MAB literature and it isn't even clear what theoretical properties hold once the MAB modifications are made for this specific domain.
- In this vein, there is limited theoretical analysis to determine how practical this approach is; how many user simulators are needed? how different do they have to be? how does one measure diversity in the simulators? There are many such open questions that aren't sufficiently addressed empirically and this gap isn't mitigated with any analytical examination (which MAB literature has a rich foundation for this type of analysis).
- MUST_uniform and MUST_adaptive are very close in final performance (although the adaptive version has better convergence properties), the MultiWoZ restaurants subset is a very specific dataset and it isn't clear if the overall results are near SoTA. Thus, there are some practical empirical questions that must be addressed -- including the general question regarding how this changes with more or fewer simulators of different characteristics.
- While less critical, there are still many typos and a general lack of narrative continuity in the writing.

Overall, the core idea is novel, the proposed approaches seem to work well empirically over single and more naive combination strategies, and the results appear promising overall. However, there remain open questions regarding if this is really a viable method as creating many sufficiently diverse user simulators isn't straightforward (and is not proven in this work). This could be addressed empirically or supported with analytical statements, but as it stands, this work has promising early results that need more practical or theoretical impact to qualify as a strong dialogue systems or MAB contribution.

Additional note: This paper was discussed with the SAC during final decisions (as there was also a 'late' change in scores). While I understand that reviewer 44kw felt their concerns were addressed, my assessment is more in line with reviewer H1xa -- that there are fundamental practical limitations of the proposed method that may limit its applicability in the general case that should be addressed for this to be a clear accept. The SAC, program chair, and myself agreed upon this decision after reading all of the reviews, rebuttals, and discussion -- and then discussing amongst ourselves.